# Asymptotic Analysis of Conditioned Stochastic Gradient Descent

**Rémi Leluc**  *remi.leluc@gmail.com*
*CMAP, École Polytechnique*
*Institut Polytechnique de Paris, Palaiseau (France)*

**François Portier**  *francois.portier@gmail.com*
*CREST, ENSAI*
*École Nationale de la Statistique et de l'Analyse de l'Information, Rennes (France)*

**Reviewed on OpenReview:** *https://openreview.net/forum?id=U4XgzRjfF1*

## Abstract

In this paper, we investigate a general class of stochastic gradient descent (SGD) algorithms, called *conditioned* SGD, based on a preconditioning of the gradient direction. Using a discrete-time approach with martingale tools, we establish under mild assumptions the weak convergence of the rescaled sequence of iterates for a broad class of conditioning matrices including stochastic first-order and second-order methods. Almost sure convergence results, which may be of independent interest, are also presented. Interestingly, the asymptotic normality result consists in a stochastic equicontinuity property so when the conditioning matrix is an estimate of the inverse Hessian, the algorithm is asymptotically optimal.

## 1 Introduction

Consider some unconstrained optimization problem of the following form:

$$\min_{\theta \in \mathbb{R}^d} \{ F(\theta) = \mathbb{E}_\xi[f(\theta, \xi)] \},$$

where $f$ is a loss function and $\xi$ is a random variable. This key methodological problem, known under the name of *stochastic programming* (Shapiro et al., 2014), includes many flagship machine learning applications such as *empirical risk minimization* (Bottou et al., 2018), *adaptive importance sampling* (Delyon & Portier, 2018) and *reinforcement learning* (Sutton & Barto, 2018). When $F$ is differentiable, a common appproach is to rely on first-order methods. However, in many scenarios and particularly in large-scale learning, the gradient of $F$ may be hard to evaluate or even intractable. Instead, a random unbiased estimate of the gradient is available at a cheap computing cost and the state-of-the-art algorithm, *stochastic gradient descent* (SGD), just moves along this estimate at each iteration. It is an iterative algorithm, simple and computationally fast, but its convergence towards the optimum is generally slow.
*Conditioned* SGD, which consists in multiplying the gradient estimate by some conditioning matrix at each iteration, can lead to better performance as shown in several recent studies ranging from natural gradient (Amari, 1998; Kakade, 2002) and stochastic second-order methods with quasi-Newton (Byrd et al., 2016) and (L)-BFGS methods (Liu & Nocedal, 1989) to diagonal scaling methods such as AdaGrad (Duchi et al., 2011), RMSProp (Tieleman et al., 2012), Adam (Kingma & Ba, 2014), AMSGrad (Reddi et al., 2018) and adaptive coordinate sampling (Wangni et al., 2018; Leluc & Portier, 2022). These conditioning techniques are based on different strategies: diagonal scaling rely on feature normalization, stochastic second-order methods are concerned with minimal variance and adaptive coordinate sampling techniques aim at taking advantage of particular data structure. Furthermore, these methods proved to be the current state-of-the-art for training machine learning models (Zhang, 2004; LeCun et al., 2012) and are implemented in widely used programming tools (Pedregosa et al., 2011; Abadi et al., 2016).

*Conditioned SGD* generalizes *standard SGD* by adding a conditioning step to refine the descent direction. Starting from $\theta_0 \in \mathbb{R}^d$, the algorithm of interest is defined by the following iteration

$$\theta_{k+1} = \theta_k - \gamma_{k+1} C_k g(\theta_k, \xi_{k+1}), \qquad k \geq 0,$$

where $g(\theta_k, \xi_{k+1})$ is some unbiased gradient valued in $\mathbb{R}^d$, $C_k \in \mathbb{R}^{d \times d}$ is called *conditioning matrix* and $(\gamma_k)_{k \geq 1}$ is a decreasing learning rate sequence. An important question, which is still open to the best of our knowledge, is to characterize the asymptotic variance of such algorithms for non-convex objective $F$ and general estimation procedure for the conditioning matrix $C_k$.

**Related work.** Seminal works around standard SGD ($C_k = I_d$) were initiated by Robbins & Monro (1951) and Kiefer et al. (1952). Since then, a large literature known as *stochastic approximation*, has developed. The almost sure convergence is studied in Robbins & Siegmund (1971) and Bertsekas & Tsitsiklis (2000); rates of convergence are investigated in Kushner & Huang (1979) and Pelletier (1998a); non-asymptotic bounds are given in Moulines & Bach (2011). The asymptotic normality can be obtained using two different approaches: a diffusion-based method is employed in Pelletier (1998b) and Benaïm (1999) whereas martingale tools are used in Sacks (1958) and Kushner & Clark (1978). We refer to Nevelson & Khas'minskiĭ (1976); Delyon (1996); Benveniste et al. (2012); Duflo (2013) for general textbooks on *stochastic approximation*.

The aforementioned results do not apply directly to *conditioned* SGD because of the presence of the matrix sequence $(C_k)_{k \geq 0}$ involving an additional source of randomness in the algorithm. Seminal papers dealing with the weak convergence of *conditioned* SGD are Venter (1967) and Fabian (1968). Within a restrictive framework (univariate case $d = 1$ and strong assumptions on the function $F$), their results are encouraging because the limiting variance of the procedure is shown to be smaller than the limiting variance of standard SGD. Venter's and Fabian's results have then been extended to more general situations (Fabian, 1973; Nevelson & Khas'minskiĭ, 1976; Wei, 1987). In Wei (1987), the framework is still restrictive not only because the random errors are assumed to be independent and identically distributed but also because the objective $F$ must satisfy their assumption (4.10) which hardly extends to objectives other than quadratic.

More recently, Bercu et al. (2020) have obtained the asymptotic normality as well as the efficiency of certain *conditioned* SGD estimates in the particular case of *logistic regression*. The previous approach has been generalized not long ago in Boyer & Godichon-Baggioni (2022) where the use of the Woodbury matrix identity is promoted to compute the Hessian inverse in the online setting. Several theoretical results, including the weak convergence of *conditioned* SGD, are obtained for convex objective functions. An alternative to *conditioning*, called *averaging*, developed by Polyak (1990) and Polyak & Juditsky (1992), allows to recover the same asymptotic variance as *conditioned SGD*. When dealing with convex objectives, the theory behind this averaging technique is a well-studied topic (Moulines & Bach, 2011; Gadat & Panloup, 2017; Dieuleveut et al., 2020; Zhu et al., 2021). However, it is inevitably associated with a large bias caused by poor initialization and requires some parameter tuning through the *burn-in* phase.

**Contributions.** The main result of this paper deals with the weak convergence of the rescaled sequence of iterates. Interestingly, our asymptotic normality result consists of the following continuity property: whenever the matrix sequence $(C_k)_{k \geq 0}$ converges to a matrix $C$ and the iterates $(\theta_k)_{k \geq 0}$ converges to a minimizer $\theta^\star$, the algorithm behaves in the same way as an oracle version in which $C$ would be used instead of $C_k$. We stress that contrary to Boyer & Godichon-Baggioni (2022), no convexity assumption is needed on the objective function and no rate of convergence is required on the sequence $(C_k)_{k \geq 0}$. This is important because, in most studies, deriving a convergence rate on $(C_k)_{k \geq 0}$ requires a specific convergence rate on the iterates $(\theta_k)_{k \geq 0}$ which, in general, is unknown at this stage of the analysis. From a more practical point of view, our main result claims that the impact of the approximation error resulting from the conditioning matrices estimation assumes a secondary role. This finding promotes the use of simple and cheap sequential algorithm to estimate the conditioning matrix which encompasses a broad spectrum of *conditioned* SGD methods, highlighting the applicability and generalizability of the obtained result. Another result of independent interest dealing with the almost sure convergence of the gradients is also provided.

In addition, for illustration purposes, we apply our results to the popular variational inference problem where one seeks to approximate a target density out of a parametric family by solving an optimization problem. In this framework, by optimizing the forward Kullback-Liebler divergence (Jerfel et al., 2021) and building

stochastic gradients relying on importance sampling schemes (Delyon & Portier, 2018), we show that the approach has some efficiency properties. For the sake of completeness, we present in appendix practical ways to compute the *conditioning* matrix $C_k$ and show that the resulting procedure satisfies the high-level conditions of our main Theorem. This yields a feasible algorithm achieving minimum variance.

To obtain these results, instead of approximating the rescaled sequence of iterates by a continuous diffusion (as for instance in Pelletier (1998b)), we rely on a discrete-time approach where the recursion scheme is directly analyzed (as for instance in Delyon (1996)). More precisely, the sequence of iterates is studied with the help of an auxiliary linear algorithm whose limiting distribution can be deduced from the central limit theorem for martingale increments (Hall & Heyde, 1980). The limiting variance is derived from a discrete time matrix-valued dynamical system algorithm. It corresponds to the solution of a Lyapunov equation involving the matrix $C$. It allows a special choice for $C$ which guarantees an optimal variance. Finally, a particular recursion is identified to examine the remaining part. By studying it on a particular event, this part is shown to be negligible.

**Outline.** Section 2 introduces the framework of standard SGD with asymptotic results. Section 3 is dedicated to *conditioned* SGD: it first presents popular optimization methods that fall in the considered framework and then presents our main results, namely the weak convergence and asymptotic optimality. Section 4 gathers practical implications of the main results for machine learning models in the framework of variational inference and Section 5 concludes the paper with a discussion of avenues for further research. Technical proofs, additional propositions and numerical experiments are available in the appendix.

## 2 Mathematical background

In this section, the mathematical background of stochastic gradient descent (SGD) methods is presented and illustrated with the help of some examples. Then, to motivate the use of *conditioning* matrices, we present a known result from Pelletier (1998b) about the weak convergence of SGD.

### 2.1 Problem setup

Consider the problem of finding a minimizer $\theta^\star \in \mathbb{R}^d$ of a function $F : \mathbb{R}^d \to \mathbb{R}$, that is,

$$\theta^\star \in \arg\min_{\theta \in \mathbb{R}^d} F(\theta).$$

In many scenarios and particularly in large scale learning, the gradient of $F$ cannot be fully computed and only a stochastic unbiased version of it is available. The SGD algorithm moves the iterate along this direction. To increase the efficiency, the random generators used to derive the unbiased gradients might evolve during the algorithm, *e.g.*, using the past iterations. To analyse such algorithms, we consider the following probabilistic setting.

**Definition 1.** *A stochastic algorithm is a sequence $(\theta_k)_{k \geq 0}$ of random variables defined on a probability space $(\Omega, \mathcal{F}, \mathbb{P})$ and valued in $\mathbb{R}^d$. Define $(\mathcal{F}_k)_{k \geq 0}$ as the natural $\sigma$-field associated to the stochastic algorithm $(\theta_k)_{k \geq 0}$, i.e., $\mathcal{F}_k = \sigma(\theta_0, \theta_1, \ldots, \theta_k)$, $k \geq 0$. A policy is a sequence of random probability measures $(P_k)_{k \geq 0}$, each defined on a measurable space $(S, \mathcal{S})$ that are adapted to $\mathcal{F}_k$.*

Given a *policy* $(P_k)_{k \geq 0}$ and a *learning rates* sequence $(\gamma_k)_{k \geq 1}$ of positive numbers, the SGD algorithm (Robbins & Monro, 1951) is defined by the update rule

$$\theta_{k+1} = \theta_k - \gamma_{k+1} g(\theta_k, \xi_{k+1}) \quad \text{with} \quad \xi_{k+1} \sim P_k, \tag{1}$$

where $g : \mathbb{R}^d \times S \to \mathbb{R}^d$ is called the gradient generator. The choice of the *policy* $(P_k)_{k \geq 0}$ in SGD is important as it can impact the convergence speed, generalization performance, and efficiency of the optimization algorithm. While most classical approaches rely on uniform sampling and mini-batch sampling, it may be more efficient to use advanced selection sampling strategy such as stratified sampling or importance sampling (see Example 1 for details). The policy $(P_k)_{k \geq 0}$ is used at each iteration to produce random gradients through the function $g$. Those gradients are assumed to be unbiased.

**Assumption 1** (Unbiased gradient)**.** *The gradient generator $g : \mathbb{R}^d \times S \to \mathbb{R}^d$ is such that for all $\theta \in \mathbb{R}^d$, $g(\theta, \cdot)$ is measurable, and we have:* $\forall k \geq 0, \quad \mathbb{E}\left[g(\theta_k, \xi_{k+1})|\mathcal{F}_k\right] = \nabla F(\theta_k).$

We emphasize three important examples covered by the developed approach. In each case, explicit ways to generate the stochastic gradient are provided.

**Example 1.** *(Empirical Risk Minimization)* Given some observed data $z_1, \ldots, z_n \in \mathbb{R}^p$ and a differentiable loss function $\ell : \mathbb{R}^d \times \mathbb{R}^p \to \mathbb{R}$, the objective function $F$ approximates the true expected risk $\mathbb{E}_z[\ell(\theta, z)]$ using its empirical counterpart $F(\theta) = n^{-1} \sum_{i=1}^n \ell(\theta, z_i)$. Classically, the gradient estimates at $\theta_k$ are given by the policy

$$g(\theta_k, \xi_{k+1}) = \nabla_\theta \ell(\theta_k, \xi_{k+1}) \quad \text{with} \quad \xi_{k+1} \sim \sum_{i=1}^n \delta_{z_i}/n.$$

Another one, more subtle, referred to as mini-batching (Gower et al., 2019), consists in generating uniformly a set of $n_k$ samples $(z_1, \ldots, z_{n_k})$ and computing the gradient as the average $n_k^{-1} \sum_{j=1}^{n_k} \nabla_\theta \ell(\theta_k, z_j)$. Note that interestingly, we allow changes of the minibatch size throughout the algorithm. Our framework also includes adaptive non-uniform sampling (Papa et al., 2015) and survey sampling (Clémençon et al., 2019), which use $P_k = \sum_{i=1}^n w_i^{(k)} \delta_{z_i}$ with $\mathcal{F}_k$-adapted weights satisfying $\sum_{i=1}^n w_i^{(k)} = 1$ for each $k \geq 0$.

**Example 2.** *(Adaptive importance sampling for variational inference)* Given a target density function $f$, which for instance might result from the posterior distribution of some observed data, and a parametric family of samplers $\{q_\theta : \theta \in \Theta\}$, the aim is to find a good approximation of $f$ out of the family of samplers. A standard choice (Jerfel et al., 2021) for the objective function is the so called *forward Kullback-Leibler* divergence given by $F(\theta) = -\int \log(q_\theta(y)/f(y)) f(y) dy$. Then in the spirit of adaptive importance sampling schemes (Delyon & Portier, 2018), gradient estimates are given by

$$g(\theta_k, \xi_{k+1}) = -\nabla_\theta \log(q_{\theta_k}(\xi_{k+1})) \frac{f(\xi_{k+1})}{q_{\theta_k}(\xi_{k+1})}, \quad \xi_{k+1} \sim q_{\theta_k}.$$

Other losses such as $\alpha$-divergence (Daudel et al., 2021) or generalized method of moment (Delyon & Portier, 2018) may also be considered depending on the problem of interest. Some applications of *conditioned* SGD algorithm to this particular framework are considered with more details in Section 4.

**Example 3.** *(Policy-gradient methods)* In reinforcement learning (Sutton & Barto, 2018), the goal of the agent is to find the best action-selection policy to maximize the expected reward. Policy-gradient methods (Baxter & Bartlett, 2001; Williams, 1992) use a parameterized policy $\{\pi_\theta : \theta \in \Theta\}$ to optimize an expected reward function $F$ given by $F(\theta) = \mathbb{E}_{\xi \sim \pi_\theta}[\mathcal{R}(\xi)]$ where $\xi$ is a trajectory including nature states and selected actions. Using the policy gradient theorem, one has $\nabla F(\theta) = \mathbb{E}_{\xi \sim \pi_\theta}[\mathcal{R}(\xi) \nabla_\theta \log \pi_\theta(\xi)]$, leading to the REINFORCE algorithm (Williams, 1992) given by

$$g(\theta_k, \xi_{k+1}) = \mathcal{R}(\xi_{k+1}) \nabla_\theta \log \pi_{\theta_k}(\xi_{k+1}), \quad \xi_{k+1} \sim \pi_{\theta_k}.$$

## 2.2 Weak convergence of SGD

This section is related to the weak convergence property of the normalized sequence of iterates $(\theta_k - \theta^\star)/\sqrt{\gamma_k}$. The working assumptions include the almost sure convergence of the sequence of iterates $(\theta_k)_{k \geq 0}$ towards a stationary point $\theta^\star$. Note that, given Assumptions 1 and 2, there exist many criteria on the objective function that give such almost sure convergence. For these results, we refer to Bertsekas & Tsitsiklis (2000); Benveniste et al. (2012); Duflo (2013). In addition to this high-level assumption of almost sure convergence, we require the following classical assumptions. Let $\mathcal{S}_d^{++}(\mathbb{R})$ denote the space of real symmetric positive definite matrices and define for all $k \geq 0$,

$$w_{k+1} = \nabla F(\theta_k) - g(\theta_k, \xi_{k+1}), \quad \Gamma_k = \mathbb{E}\left[w_{k+1} w_{k+1}^\top | \mathcal{F}_k\right].$$

The learning rates sequence $(\gamma_k)_{k \geq 1}$ should decay to eventually anneal the noise but not too fast so that the iterates $(\theta_k)_{k \geq 0}$ can reach interesting places in a finite time.

**Assumption 2** (Learning rates). *The sequence of step-size is $\gamma_k = \alpha k^{-\beta}$ with $\beta \in (1/2, 1]$.*

This classical form of the step-size ensures theoretical convergence guarantee through the Robbins-Monro condition: $\sum_k \gamma_k = \infty, \sum_k \gamma_k^2 < \infty$. However, note that in practice, the choice of learning rate is often determined through experimentation and fine-tuning to achieve the best performance on the given task.

**Assumption 3** (Hessian). *The Hessian matrix at stationary point is positive definite, i.e., $H = \nabla^2 F(\theta^\star) \in \mathcal{S}_d^{++}(\mathbb{R})$ and the mapping $\theta \mapsto \nabla^2 F(\theta)$ is continuous at $\theta^\star$.*

The positive definiteness of the Hessian matrix provides stability and robustness guarantees in the optimization process. It ensures that small perturbations or noise in the objective function or the training data do not significantly affect the convergence behavior. The positive curvature helps in confining the optimization trajectory near the minimum and prevents it from getting trapped in flat regions or saddle points.

The noise sequence $(w_k)_{k \geq 1}$ defines a sequence of conditional covariance matrices $(\Gamma_k)_{k \geq 1}$ that is assumed to converge so that one can identify the limiting covariance $\Gamma = \mathbb{E}[g(\theta^\star, \xi)g(\theta^\star, \xi)^\top]$.

**Assumption 4** (Covariance matrix). *There exists $\Gamma \in \mathcal{S}_d^{++}(\mathbb{R})$ such that $\Gamma_k \overset{k \to +\infty}{\longrightarrow} \Gamma$ a.s.*

Finally, in order to derive a central limit theorem for the iterates of the algorithm, there is an extra need for stability which is synonymous with a uniform bound on the noise around the minimizer.

**Assumption 5** (Lyapunov bound). *There exist $\delta, \varepsilon > 0$ such that:*

$$\sup_{k \geq 0} \mathbb{E}[\|w_{k+1}\|_2^{2+\delta}|\mathcal{F}_k]\mathbb{1}_{\{\|\theta_k - \theta^\star\| \leq \varepsilon\}} < \infty \quad a.s.$$

Note that all these assumptions are stated in the spirit of Pelletier (1998b) making them mild and general. In particular, Assumptions 4 and 5 are similar to (A1.2) in Pelletier (1998b). More precisely, Assumption 4 is needed to identify the limiting distribution while Assumption 5 is a stability condition often referred to as the Lyapunov condition. This last condition is technical but not that strong as it is similar to the Lindeberg's condition which is necessary (Hall & Heyde, 1980) for tightness. The following result can be either derived from (Pelletier, 1998b, Theorem 1) or as a direct corollary of our main result, Theorem 2, given in Section 3.2.

**Theorem 1** (Weak convergence of SGD). *Let $(\theta_k)_{k \geq 0}$ be obtained by the SGD rule (1). Suppose that Assumptions 1, 2, 3, 4, 5 are fulfilled and that $\theta_k \to \theta^\star$ almost surely. If moreover, $(H - \zeta I)$ is positive definite with $\zeta = \mathbb{1}_{\{\beta=1\}}/2\alpha$, it holds that*

$$\frac{1}{\sqrt{\gamma_k}}(\theta_k - \theta^\star) \rightsquigarrow \mathcal{N}(0, \Sigma), \qquad as\ k \to \infty$$

*where $\Sigma$ satisfies the Lyapunov equation: $(H - \zeta I_d)\Sigma + \Sigma(H - \zeta I_d)^\top = \Gamma$.*

Several remarks are to be explored. Since $\Gamma$ and $(H - \zeta I)$ are positive definite matrices, there exists a unique solution $\Sigma$ to the Lyapunov equation $(H - \zeta I_d)\Sigma + \Sigma(H - \zeta I_d)^\top = \Gamma$ given by $\Sigma = \int_0^{+\infty} \exp[-t(H - \zeta I_d)]\Gamma \exp[-t(H - \zeta I_d)^\top]dt$. Second, the previous result can be expressed as $k^{\beta/2}(\theta_k - \theta^\star) \rightsquigarrow \mathcal{N}(0, \alpha\Sigma)$. Hence, the fastest rate of convergence is obtained when $\beta = 1$ for which we recover the classical $1/\sqrt{k}$-rate of a Monte Carlo estimate. In this case, the coefficient $\alpha$ should be chosen large enough to ensure the convergence through the condition $H - I_d/(2\alpha) \succ 0$, but also such that the covariance matrix $\alpha\Sigma$ is small. The choice of $\alpha$ is discussed in the next section and should be replaced with a matrix gain.

## 3 The asymptotics of conditioned stochastic gradient descent

This Section first presents practical optimization schemes that fall in the framework of *conditioned* SGD. Then it contains our main results, namely the weak convergence and asymptotic optimality. Another result of independent interest dealing with the almost sure convergence of the gradients and the iterates is also provided.

### 3.1 Framework and Examples

We introduce the general framework of *conditioned* SGD as an extension of the standard SGD presented in Section 2. It is defined by the following update rule, for $k \geq 0$,

$$\theta_{k+1} = \theta_k - \gamma_{k+1} C_k g(\theta_k, \xi_{k+1}), \tag{2}$$

where the *conditioning matrix* $C_k \in \mathcal{S}_d^{++}(\mathbb{R})$ is a $\mathcal{F}_k$-measurable real symmetric positive definite matrix so that the search direction always points to a descent direction. In convex optimization, inverse of the Hessian is a popular choice but *(1)* it may be hard to compute, *(2)* it is not always positive definite and *(3)* it may increase the noise of SGD especially when the Hessian is ill-conditioned.

**Quasi-Newton.** These methods build approximations of the Hessian $C_k \approx \nabla^2 F(\theta_k)^{-1}$ with gradient-only information, and are applicable for convex and nonconvex problems. For scalability issue, variants with limited memory are the most used in practice (Liu & Nocedal, 1989). Following Newton's method idea with the secant equation, the update rule is based on pairs $(s_k, y_k)$ tracking the differences of iterates and stochastic gradients, *i.e.*, $s_k = \theta_{k+1} - \theta_k$ and $y_k = g(\theta_{k+1}, \xi_{k+1}) - g(\theta_k, \xi_{k+1})$. Let $\rho_k = 1/(s_k^\top y_k)$ then the Hessian updates are

$$C_{k+1} = (I - \rho_k y_k s_k^\top)^\top C_k (I - \rho_k y_k s_k^\top) + \rho_k s_k s_k^\top.$$

In the deterministic setting, the BFGS update formula above is well-defined as long as $s_k^\top y_k > 0$. Such condition preserves positive definite approximations and may be obtained in the stochastic setting by replacing the Hessian matrix with a Gauss-Newton approximation and using regularization.

**Adaptive methods and Diagonal scalings.** These methods adapt locally to the structure of the optimization problem by setting $C_k$ as a function of past stochastic gradients. General adaptive methods differ in the construction of the *conditioning* matrix and whether or not they add a momentum term. Using different representations such as dense or sparse conditioners also modify the properties of the underlying algorithm. For instance, the optimizers Adam and RMSProp maintain an exponential moving average of past stochastic gradients with a factor $\tau \in (0, 1)$ but fail to guarantee $C_{k+1} \preceq C_k$. Such behaviour can lead to large fluctuations and prevent convergence of the iterates. Instead, AdaGrad and AMSGrad ensure the monotonicity $C_{k+1} \preceq C_k$.

| Optimizer | Gradient matrix $G_{k+1}$ | m |
|---|---|---|
| AdaFull | $G_k + g_k g_k^\top$ | 0 |
| AdaNorm | $G_k + \|g_k\|_2^2$ | 0 |
| AdaDiag | $G_k + diag(g_k g_k^\top)$ | 0 |
| RMSProp | $\tau G_k + (1 - \tau) diag(g_k g_k^\top)$ | 0 |
| Adam | $[\tau G_k + (1 - \tau) diag(g_k g_k^\top)]/(1 - \tau^k)$ | m |
| AMSGrad | $[\tau G_k + (1 - \tau) diag(g_k g_k^\top)]/(1 - \tau^k)$ | m |

Table 1: Adaptive Gradient Methods.

Denote by $g_k = g(\theta_k, \xi_{k+1})$ and $m \in [0, 1)$ a momentum parameter. General adaptive gradient methods are defined by: $\theta_{k+1} = \theta_k - \gamma_{k+1} C_k \hat{g}_k$, $\quad \hat{g}_k = m\hat{g}_{k-1} + (1 - m)g_k$. Different optimizers are summarized in Table 1 above. They all rely on a gradient matrix $G_k$ which accumulates the information of stochastic gradients. The *conditioning* matrix is equal to $C_k = G_k^{-1/2}$ except for AMSGrad which uses $C_k = \max\{C_{k-1}; G_k^{-1/2}\}$. Starting from $G_0 = \delta I$ with $\delta > 0$, $G_{k+1}$ is updated either in a dense or sparse (diagonal) manner or using an exponential moving average. Note that *conditioned SGD* methods also include schemes with general estimation of the matrix $C_k$ such as Hessian sketching (Gower et al., 2016) or Jacobian sketching (Gower et al., 2021).

A common assumption made in the literature of adaptive methods is that *conditioning* matrices are well-behaved in the sense that their eigenvalues are bounded in a fixed interval. This property is easy to check for diagonal matrices and can always be implemented in practice using projection.

### 3.2 Main result

Similarly to standard SGD, it is interesting to search for an appropriate rescaled process to obtain some convergence rate and asymptotic normality results. In fact the only additional assumption needed, compared to SGD, is the almost sure convergence of the sequence $(C_k)_{k\geq0}$. This makes Theorem 1 a particular case of the following Theorem which is the main result of the paper (the proof is given in Appendix A.1).

**Theorem 2** (Weak convergence of Conditioned SGD). *Let $(\theta_k)_{k\geq0}$ be obtained by conditioned SGD* (2). *Suppose that Assumptions 1, 2, 3, 4, 5 are fulfilled and that $\theta_k \to \theta^\star$ almost surely. If moreover, $C_k \to C \in \mathcal{S}_d^{++}(\mathbb{R})$ almost surely and all the eigenvalues of $(CH - \zeta I)$ are positive with $\zeta = \mathbb{1}_{\{\beta=1\}}/2\alpha$, it holds that*

$$\frac{1}{\sqrt{\gamma_k}}(\theta_k - \theta^\star) \rightsquigarrow \mathcal{N}(0, \Sigma_C), \qquad as\ k \to \infty,$$

*where $\Sigma_C$ satisfies:* $(CH - \zeta I_d)\,\Sigma_C + \Sigma_C\,(CH - \zeta I_d)^\top = C\Gamma C^\top.$

**Sketch of the proof.** The idea of the proof is to rely on the following bias-variance decomposition. Remark that the difference $\Delta_k = \theta_k - \theta^\star$ is subjected to the iteration:

$$\Delta_{k+1} = \Delta_k - \gamma_{k+1}C_k\nabla F(\theta_k) + \gamma_{k+1}C_k w_{k+1}, \qquad k \geq 0.$$

In a similar spirit as in Delyon (1996), we use the Taylor approximation $\nabla F(\theta_k) = \nabla F(\theta^\star) + H(\theta_k - \theta^\star) + o(\theta_k - \theta^\star) \simeq H(\theta_k - \theta^\star)$ to define the following auxiliary linear stochastic algorithm which carries the same variance as the main algorithm,

$$\widetilde{\Delta}_{k+1} = \widetilde{\Delta}_k - \gamma_{k+1}K\widetilde{\Delta}_k + \gamma_{k+1}C_k w_{k+1}, \qquad k \geq 1,$$

where $K = CH$. As a first step we establish the weak convergence of $\widetilde{\Delta}_{k+1}$ using discrete martingale tools. Note that the analysis is made possible because the matrix $K$ is fixed along this algorithm. As a second step, we prove that the difference $(\Delta_k - \widetilde{\Delta}_k)$, which represents some bias term, is negligible.

**Comparison with previous works.** Theorem 2 stated above is comparable to Theorem 1 given in Pelletier (1998b). However, our result on the weak convergence cannot be recovered from the one of Pelletier (1998b) due to their Assumption (A1.2) about convergence rates. Indeed, this assumption would require that the sequence $(C_k)_{k\geq0}$ converges towards $C$ faster than $\sqrt{\gamma_k}$. This condition is either hardly meet in practice or difficult to check. Unlike this prior work, our result only requires the almost sure convergence of the sequence $(C_k)_{k\geq0}$. In a more restrictive setting of convex objective and online learning framework, *i.e.* in which data becomes available in a sequential order, another way to obtain the weak convergence of the rescaled sequence of iterates $(\theta_k - \theta^\star)/\sqrt{\gamma_k}$ is to rely on the results of Boyer & Godichon-Baggioni (2022). However, once again, their work rely on a particular convergence rate for the matrix sequence $(C_k)_{k\geq0}$. This implies the derivation of an additional result on the almost sure convergence rate of the iterates. To overcome all these issues, we show in Appendix B that our conditions on the matrices $C_k$ are easily satisfied in common situations.

### 3.3 Asymptotic optimality of Conditioned SGD

The best *conditioning* matrix $C$ that could be chosen regarding the asymptotic variance is specified in the next proposition whose proof is given in the supplementary material (Appendix C.3).

**Proposition 1** (Optimal choice). *The choice $C^\star = H^{-1}$ is optimal in the sense that $\Sigma_{C^*} \preceq \Sigma_C$ for all $C \in \mathcal{C}_H$. Moreover, we have $\Sigma_{C^\star} = H^{-1}\Gamma H^{-1}$.*

Another remarkable result, which directly follows from the Theorem 2 is now stated as a corollary.

**Corollary 1** (Asymptotic optimality). *Under the assumptions of Theorem 2, if $\gamma_k = 1/k$ and $C = H^{-1}$, then*

$$\sqrt{k}(\theta_k - \theta^\star) \rightsquigarrow \mathcal{N}(0, H^{-1}\Gamma H^{-1}), \qquad as\ k \to \infty.$$

*Moreover, let $(Z_1, \ldots, Z_d) \sim \mathcal{N}(0, I_d)$ and $(\lambda_k)_{k=1,\ldots,d}$ be the eigenvalues of the matrix $H^{-1/2}\Gamma H^{-1/2}$, we have the convergence in distribution:*

$$k(F(\theta_k) - F(\theta^\star)) \rightsquigarrow \sum_{k=1}^{d} \lambda_k Z_k^2, \quad as \ k \to \infty.$$

This result shows the success of the proposed approach as the asymptotic variance is the optimal one. It provides the user a practical choice for the sequence of rate, $\gamma_k = 1/k$ and also removes the assumption that $2\alpha H \succ I_d$ which is usually needed in SGD (see Theorem 1). Concerning the almost sure convergence of the *conditioning* matrices, we provide in Appendix B an explicit way to ensure that $C_k \to H^{-1}$.

The above statement also provides insights about the convergence speed. It claims that the convergence rate of $F(\theta_k)$ towards the optimum $F(\theta^\star)$, in $1/k$, is faster than the convergence rate of the iterates, in $1/\sqrt{k}$. Another important feature, which is a consequence of Proposition 1, is that the eigenvalues $(\lambda_k)_{k=1,\ldots,d}$ that appear in the limiting distribution are the smallest ones among all the other possible version of *conditioned* SGD (defined by the matrix $C$).

### 3.4 Convergence of the iterates $(\theta_k)$ of Conditioned SGD

To apply both Theorem 2 and Corollary 1, it remains to check the almost sure convergence of the iterates. In a non-convex setting, the iterates of stochastic first-order methods can only reach local optima, *i.e.* the iterates are expected to converge to the following set $\mathcal{S} = \{\theta \in \mathbb{R}^d : \nabla F(\theta) = 0\}$. Going in this direction, we first prove the almost sure convergence of the gradients towards zero for general *conditioned* SGD methods under mild assumptions. This theoretical result may be of independent interest. Under a condition on $\mathcal{S}$, one may uniquely identify a limit point $\theta^\star$ and consider the event $\{\theta_k \to \theta^\star\}$ which is needed for the weak convergence results. The next analysis is based on classical assumptions which are used in the literature to obtain the convergence of standard SGD.

**Assumption 6** (L-smooth). $\exists L > 0 : \forall \theta, \eta \in \mathbb{R}^d, \quad \|\nabla F(\theta) - \nabla F(\eta)\|_2 \le L\|\theta - \eta\|_2$.

**Assumption 7** (Lower bound). $\exists F^\star \in \mathbb{R} : \forall \theta \in \mathbb{R}^d, F^\star \le F(\theta)$.

To handle the noise of the stochastic estimates, we consider a weak growth condition, related to the notion of *expected smoothness* as introduced in Gower et al. (2019) (see also Gazagnadou et al. (2019); Gower et al. (2021)). In particular, we extend the condition of Gower et al. (2019) to our general context in which the sampling distributions are allowed to change along the algorithm.

**Assumption 8** (Growth condition). *With probability $1$, there exist $0 \le \mathcal{L}, \sigma^2 < \infty$ such that for all $\theta \in \mathbb{R}^d, k \in \mathbb{N}, \quad \mathbb{E}\left[\|g(\theta, \xi_{k+1})\|_2^2 | \mathcal{F}_k\right] \le 2\mathcal{L}(F(\theta) - F^\star) + \sigma^2$.*

This almost-sure bound on the stochastic noise $\mathbb{E}\left[\|g(\theta, \xi_k)\|_2^2 | \mathcal{F}_{k-1}\right]$ is key in the analysis of the *conditioned* SGD algorithm. This weak growth condition on the stochastic noise is general and can be achieved in practice with a general Lemma available in the supplement (Appendix C.4). Note that Assumption 8, often referred to as a *growth condition*, is mild since it allows the noise to be large when the iterate is far away from the optimal point. In that aspect, it contrasts with uniform bounds of the form $\mathbb{E}\left[\|g(\theta_k, \xi_{k+1})\|_2^2 | \mathcal{F}_k\right] \le \sigma^2$ for some deterministic $\sigma^2 > 0$ (see Nemirovski et al. (2009); Nemirovski & Yudin (1983); Shalev-Shwartz et al. (2011)). Observe that such uniform bound is recovered by taking $\mathcal{L} = 0$ in Assumption 8 but cannot hold when the objective function $F$ is strongly convex (Nguyen et al., 2018). Besides, fast convergence rates have been derived in Schmidt & Roux (2013) under the *strong-growth condition*: $\mathbb{E}[\|g(\theta, \xi_{k+1})\|_2^2 | \mathcal{F}_k] \le M\|\nabla F(\theta)\|_2^2$ for some $M > 0$. Similarly to our growth condition, Bertsekas & Tsitsiklis (2000) and Bottou et al. (2018) performed an analysis under the condition $\mathbb{E}[\|g(\theta, \xi_{k+1})\|_2^2 | \mathcal{F}_k] \le M\|\nabla F(\theta)\|_2^2 + \sigma^2$ for $M, \sigma^2 > 0$. Under Assumptions 6 and 7, we have $\|\nabla F(\theta)\|_2^2 \le 2L(F(\theta) - F(\theta^\star))$ (Gower et al., 2019, Proposition A.1) so our growth condition is less restrictive. If $F$ satisfies the Polyak-Lojasiewicz condition (Karimi et al., 2016), then our growth condition becomes a bit stronger. Another weak growth condition has been used for a non-asymptotic study in Moulines & Bach (2011). The success of *conditioned* SGD relies on the following extended Robbins-Monro condition which ensures a control on the eigenvalues of the *conditioning* matrices.

**Assumption 9** (Eigenvalues). *Let $(\mu_k)_{k\geq 1}$ and $(\nu_k)_{k\geq 1}$ be positive sequences such that:*
$$\forall k \geq 1, \mu_k I_d \preceq C_{k-1} \preceq \nu_k I_d; \quad \sum_k \gamma_k \nu_k = +\infty; \quad \sum_k (\gamma_k \nu_k)^2 < +\infty; \quad \limsup_k \nu_k/\mu_k < \infty \ a.s.$$

The last condition deals with the ratio $(\nu_k/\mu_k)$ which may be seen as a conditioned number and ensures that the matrices $C_k$ are well-conditioned. The following Theorem reveals that all these assumptions are sufficient to ensure the almost sure convergence of the gradients towards zero.

**Theorem 3** (Almost sure convergence). *Suppose that Assumptions 1, 6, 7, 8, 9 are fulfilled. Then $(\theta_k)_{k\geq 0}$ obtained by conditioned SGD (2) satisfies $\nabla F(\theta_k) \to 0$ as $k \to \infty$ a.s.*

Other convergence results concerning the sequence of iterates towards global minimizers may be obtained by considering stronger assumptions such as convexity or that $F$ is coercive and the level sets of stationary point $\mathcal{S} \cap \{\theta, F(\theta) = y\}$ are locally finite for every $y \in \mathbb{R}^d$ (see Gadat et al. (2018)). In our analysis, the proof of Theorem 3 reveals that $\theta_{k+1} - \theta_k \to 0$ in $L^2$ and almost surely. Thus, as soon as the stationary points are isolated, the sequence of iterates will converge towards a unique stationary point $\theta^\star \in \mathbb{R}^d$. This result is stated in the next Corollary.

**Corollary 2** (Almost sure convergence). *Under the assumptions of Theorem 3, assume that $F$ is coercive and let $(\theta_k)_{k\geq 0}$ be the sequence of iterates obtained by the conditioned SGD (2), then $d(\theta_k, \mathcal{S}) \to 0$ as $k \to \infty$. In particular, if $\mathcal{S}$ is a finite set, $(\theta_k)$ converges to some $\theta^\star \in \mathcal{S}$.*

## 4 Asymptotic optimality in Adaptive importance sampling

The aim of this section is to demonstrate that statistical efficiency can be ensured in variational inference problems through the combination of adaptive importance sampling and *conditioned* SGD. While well-known results regarding the asymptotic optimality of maximum likelihood estimates (MLE) obtained from *conditioned* SGD (see for instance Amari (1998) or Bercu et al. (2020)) are initially revisited, attention is subsequently shifted towards the variational inference topic relying on adaptive sampling schemes methods. A novel result is then presented, asserting that even within this challenging framework, *conditioned* SGD allows for the recovery of a certain statistical efficiency.

### 4.1 Maximum likelihood estimation

Assume that $(X_k)_{k\geq 1}$ is an independent sequence of random variables with distribution $q^\star$. Consider a parametric family $\{q_\theta : \theta \in \Theta\}$ from which we aim to obtain an estimate of $q^\star$. We further assume that the model is well-specified, *i.e.* $q^\star = q_{\theta^\star}$ for some $\theta^\star \in \Theta$. The MLE is given by

$$\hat{\theta}_n \in \arg\max_{\theta \in \Theta} \sum_{i=1}^{n} \log(q_\theta(X_i)).$$

Under suitable condition (van der Vaart, 1998), it is well known that $\hat{\theta}_n$ is efficient, meaning it is asymptotically unbiased and has the smallest achievable variance. The Cramer-Rao bound is given by the inverse of the Fisher information matrix, denoted by $\mathcal{I}^{-1}$ and defined as

$$\mathcal{I} = \int \nabla_\theta \log(q_{\theta^\star}) \nabla_\theta \log(q_{\theta^\star})^\top q_{\theta^\star} d\lambda.$$

Unfortunately, the estimate $\hat{\theta}_n$ is often unknown in closed-form, requiring the use of a sequential procedure for approximation. This raises the further question of whether the estimate obtained through the sequential procedure achieves the efficiency bound. When using standard SGD without conditioning, the update rule is $\theta_{k+1} = \theta_k - \gamma_{k+1} \nabla \log(q_{\theta_k}(X_{k+1}))$. However, the optimal variance bound is not achieved in this case. To recover efficiency, one can rely on *conditioned* SGD, incorporating a conditioning matrix that estimates the inverse of the Hessian. In light of the definition of the Fisher information matrix, the conditioning matrix can be estimated iteratively using at each step a new sample $X_{k+1}$ as follows

$$\mathcal{I}_{k+1} = (1 - \gamma_{k+1})\mathcal{I}_k + \gamma_{k+1} \nabla \log(q_{\theta_k}(X_k)) \nabla_\theta \log(q_{\theta_k}(X_k))^\top$$

and then relying on the CSGD algorithm with update rule $\theta_{k+1} = \theta_k - \gamma_{k+1}\mathcal{I}_k^{-1}\nabla \log(q_{\theta_k}(X_{k+1}))$. As a consequence of Theorem 2, under stipulated assumptions, one can recover the optimal bound $\mathcal{I}^{-1}$ as the asymptotic variance of $\sqrt{k}(\theta_k - \theta^*)$.

## 4.2 Adaptive importance sampling

Consider the variational inference problem where the aim is to approximate a target distribution $q^\star = q_{\theta^\star}$ based on a family of density $\{q_\theta : \theta \in \Theta\}$. Unlike the previous statistical framework, one does not have access to random variables distributed according to $q^\star$. Instead, one can usually evaluate the target function $q^\star$. More background about this type of problem might be found in Zhang et al. (2018). In the following, we show that *conditioned* SGD methods allow to achieve the same variance as the optimal variance described in the previous statistical setting. To the best of our knowledge, this result is novel and has potential implications in variational inference problems using forward KL (or $\alpha$-divergence) as described in Jerfel et al. (2021) and Section 5.2 in Zhang et al. (2018). Consider the objective function defined as the Kullback-Liebler divergence between a sampler $q_\theta$ and the target distribution $q^\star$, *i.e.*,

$$F(\theta) = -\int \log(q_\theta/q^\star)q^\star d\lambda.$$

Under regularity conditions, the gradient and Hessian are respectively written as $\nabla_\theta F(\theta) = -\mathbb{E}_{q^\star}[\nabla_\theta \log(q_\theta)]$ and $\nabla_\theta^2 F(\theta) = -\mathbb{E}_{q^\star}[\nabla_\theta^2 \log(q_\theta)]$. Stochastic gradients can be defined using adaptive importance sampling-based estimate as in Delyon & Portier (2018). Given the current iterate $\theta_k$, one needs to generate $X_{k+1}$ from $q_{\theta_k}$ and compute the (unbiased) stochastic gradient $g_{k+1} = v_{k+1}\nabla_{k+1}$ where $v_{k+1} = q^\star(X_{k+1})/q_{\theta_k}(X_{k+1})$ and $\nabla_{k+1} = \nabla_\theta \log(q_{\theta_k}(X_{k+1}))$. Based on our almost sure convergence result, one can obtain that $\theta_k \to \theta^*$ and then deduce

$$\Gamma = \lim_{k\to\infty} \mathbb{E}[g_{k+1}g_{k+1}^\top] = \lim_{k\to\infty} \int \nabla_\theta \log(q_{\theta_k})\nabla_\theta \log(q_{\theta_k})^\top \frac{q^{\star 2}}{q_{\theta_k}}d\lambda = \mathcal{I},$$

where the value $\mathcal{I}$ for the limit comes from replacing $q_{\theta_k}$ by its limit $q_{\theta^*} = q^*$. The choice of the *conditioning* matrix $C_k$ may be done using an auxiliary algorithm of the following form

$$C_{k+1} = (1 - \gamma_{k+1})C_k + \gamma_{k+1}v_{k+1}\nabla_{k+1}\nabla_{k+1}^\top.$$

It can be shown that the sequence of *conditioning* matrices $(C_k)$ converges to $\mathcal{I}$. Thus, Theorem 2 implies that *conditioned* SGD is efficient in this framework as it matches the lower bound of the previous less restrictive statistical framework in which $q_k = q^\star$. Similar computations, left for future work, may be performed to investigate if the same optimal variance can be achieved with more general similarity measures such as $\alpha$-divergences (Daudel et al., 2021).

## 5 Conclusion and Discussion

We derived an asymptotic theory for *Conditioned* SGD methods in a general non-convex setting. Compared to standard SGD methods, the only additional assumption required to obtain the weak convergence is the almost sure convergence of the *conditioning* matrices. The use of appropriate *conditioning* matrices with the help of Hessian estimates is the key to achieve asymptotic optimality. While our study focuses on the weak convergence of the rescaled sequence of iterates - an appropriate tool to deal with efficiency issues since algorithms can be easily compared through their asymptotic variances - it would be interesting to complement our asymptotic results with concentration inequalities. This research direction, left for future work, may be done at the cost of extra assumptions, *e.g.*, strong convexity of the objective function combined with bounded gradients. Furthermore, by using some recent results on the behavior of adaptive gradient methods in non-convex settings (Daneshmand et al., 2018; Staib et al., 2019; Antonakopoulos et al., 2022), another research direction would be to extend the current weak convergence analysis to edge cases where the objective function possesses saddle points.

From a practical standpoint, the approach proposed in Appendix B may not be computationally optimal as it requires eigenvalue decomposition. However, *conditioned* SGD methods and especially stochastic second-order methods do not actually require the full computation of a matrix decomposition but rely on matrix-vector products which may be performed in $O(d^2)$ operations. Futhermore, using low-rank approximation

with BFGS algorithm (Broyden, 1970; Fletcher, 1970; Goldfarb, 1970; Shanno, 1970) and its variant L-BFGS (Liu & Nocedal, 1989), those algorithms approximately invert Hessian matrices in $O(d)$ operations. More recently, this technique was extended to the online learning framework (Schraudolph et al., 2007) and a purely stochastic setting (Moritz et al., 2016). Similarly, the different adaptive optimizers presented in Section 3.1 are concerned with both fast computations and high precision. Designing an efficient *conditioned* SGD algorithm involves a careful trade-off between the low-memory storage of the scaling matrix representation $C_k$ and the quality of its approximation of either the inverse Hessian $\nabla^2 F(\theta^\star)^{-1}$ or the information brought in by the underlying geometry of the problem.

## Acknowledgments

The authors are grateful to the Associate Editor and three anonymous Reviewers for their many valuable comments and interesting suggestions.

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

# Appendix: Asymptotic Analysis of Conditioned Stochastic Gradient Descent

Appendix A contains the mathematical proofs of the main results while Appendix B is dedicated to a practical procedure and numerical experiments for illustration purposes. Appendix C gathers some technical auxiliary results and additional propositions.

## A Proofs of main results

### A.1 Proof of the weak convergence (Theorem 2)

For any matrix $A \in \mathbb{R}^{d \times d}$, we denote by $\|A\| = \max_{\|u\|_2=1} \|Au\|_2$ the operator norm associated to the Euclidian norm and by $\rho(A)$ the spectral radius of $A$, *i.e.*, $\rho(A) = \max\{|\lambda_1|, \ldots, |\lambda_n|\}$ where $\lambda_1, \ldots, \lambda_n$ are the eigenvalues of $A$. We also introduce $\lambda_{\min}(A) = \min\{|\lambda_1|, \ldots, |\lambda_n|\}$. Note that when $A$ is symmetric $\|A\| = \rho(A)$ and recall that the spectral radius is a (submultiplicative) norm on the real linear space of symmetric matrices.

**Structure of the proof.**

In virtue of Assumption 5, there exist $\delta, \varepsilon > 0$ such that almost surely

$$\sup_{k \geq 0} \mathbb{E}[\|w_{k+1}\|_2^{2+\delta} | \mathcal{F}_k] \mathbb{1}_{\{\|\theta_k - \theta^\star\|_2 \leq \varepsilon\}} < \infty. \tag{3}$$

An important event in the following is

$$\mathcal{A}_k = \{\|\theta_k - \theta^\star\|_2 \leq \varepsilon, \ \|C_k\| < 2\|C\|, \ \|\Gamma_k\| \leq 2\|\Gamma\|\}.$$

By assumption, this event has probability going to 1.

Introduce the difference

$$\Delta_k = \theta_k - \theta^\star,$$

and remark that $\Delta_k$ is subjected to the iteration:

$$\Delta_0 = \theta_0 - \theta^\star,$$
$$\Delta_{k+1} = \Delta_k - \gamma_{k+1} C_k \nabla F(\theta_k) + \gamma_{k+1} C_k w_{k+1}, \qquad k \geq 0,$$

with $w_{k+1} = \nabla F(\theta_k) - g(\theta_k, \xi_{k+1})$. We have by assumption that $C_k \to C$ almost surely and we can define $K = \lim_{k\to\infty} C_k H = CH$. The proof relies on the introduction of an auxiliary stochastic algorithm which follows the iteration:

$$\widetilde{\Delta}_0 = \theta_0 - \theta^\star$$
$$\widetilde{\Delta}_{k+1} = \widetilde{\Delta}_k - \gamma_{k+1} K \widetilde{\Delta}_k + \gamma_{k+1} C_k w_{k+1} \mathbb{1}_{\mathcal{A}_k}, \qquad k \geq 0$$

The previous algorithm is a linear approximation of the algorithm that defines $\Delta_k$ in the sense that $\nabla F(\theta_k) = \nabla F(\theta^\star) + H(\theta_k - \theta^\star) + o(\theta_k - \theta^\star) \simeq H(\theta_k - \theta^\star)$ has been linearly expanded around $\theta^\star$. Writing

$$\Delta_k = \widetilde{\Delta}_k + (\Delta_k - \widetilde{\Delta}_k),$$

and invoking the Slutsky lemma, the proof will be complete as soon as we obtain that

$$\gamma_k^{-1/2} \widetilde{\Delta}_k \rightsquigarrow \mathcal{N}(0, \Sigma), \tag{4}$$
$$(\Delta_k - \widetilde{\Delta}_k) = o_{\mathbb{P}}(\gamma_k^{1/2}). \tag{5}$$

Denote by $\sqrt{H}$ the positive square root of the real symmetric positive definite matrix $H$ and consider the transformation $\Theta_k = \sqrt{H}\widetilde{\Delta}_k$ which satisfies

$$\Theta_0 = \sqrt{H}\widetilde{\Delta}_0$$
$$\Theta_{k+1} = \Theta_k - \gamma_{k+1}\widetilde{K}\Theta_k + \gamma_{k+1}\sqrt{H}C_k w_{k+1}\mathbb{1}_{\mathcal{A}_k}, \qquad k \geq 1,$$

where $\widetilde{K} = \sqrt{H}C\sqrt{H} \in \mathcal{S}_d^{++}(\mathbb{R})$ is a real symmetric positive definite matrix. The sequence $(\Theta_k)_{k\geq 0}$ is easier to study than $\widetilde{\Delta}_k$ because contrary to $\widetilde{K}$, the matrix $K = CH$ is not symmetric in general unless $C$ and $H$ commute. In view of Assumption 3, the eigenvalues of $\widetilde{K}$ are real and positive. Denote by $\lambda_m$ (*resp.* $\lambda_M$) the smallest (*resp.* the largest) eigenvalue of $\widetilde{K}$, *i.e.*,

$$\lambda_m = \lambda_{\min}(\widetilde{K}), \quad \lambda_M = \lambda_{\max}(\widetilde{K}).$$

Because $CH$ is similar to $\widetilde{K}$, they share the same eigenvalues. Since by assumption, the eigenvalues of $(CH - \zeta I_d)$ are positive, we have $2\alpha\lambda_m > \mathbb{1}_{\{\beta=1\}}$. For all $k \geq 1$, introduce the real symmetric matrix $A_k = I - \gamma_k \widetilde{K}$. Observe that all these matrices commute, i.e., for any $i, j \geq 0$, we have $A_i A_j = A_j A_i$. For any $k, n \geq 0$, denote the matrices product

$$\begin{cases} \Pi_{n,k} &= A_n \ldots A_{k+1} \text{ if } k < n \\ \Pi_{n,k} &= I_d \text{ if } k \geq n, \Pi_n = \Pi_{n,0} \end{cases}$$

Since the matrices $A_k$ commute, we have $\Pi_{n,k}^\top = \Pi_{n,k}$ is also real symmetric.

**Step 1. Proof of Equation** (4).
The random process $(\Theta_k)_{k\geq 0}$ follows the recursion equation

$$\Theta_k = A_k \Theta_{k-1} + \gamma_k \sqrt{H} C_{k-1} w_k \mathbb{1}_{\mathcal{A}_{k-1}}.$$

We have by induction

$$\Theta_n = \Pi_n \Theta_0 + \sum_{k=1}^n \gamma_k \Pi_{n,k} \sqrt{H} C_{k-1} w_k \mathbb{1}_{\mathcal{A}_{k-1}},$$

and the rescaled process is equal to

$$\frac{\Theta_n}{\sqrt{\gamma_n}} = \underbrace{\frac{\Pi_n}{\sqrt{\gamma_n}} \Theta_0}_{\textit{initial error } Y_n} + \underbrace{\sum_{k=1}^n \frac{\gamma_k}{\sqrt{\gamma_n}} \Pi_{n,k} \sqrt{H} C_{k-1} w_k \mathbb{1}_{\mathcal{A}_{k-1}}}_{\textit{sampling error } M_n}.$$

**Bound on the initial error.**
Define $\tau_n = \sum_{k=1}^n \gamma_k$ the partial sum of the learning rates. Since $\Pi_n$ is symmetric, we have $\rho(\Pi_n\Theta_0) \leq \rho(\Pi_n)\|\Theta_0\|_2$. In view of Lemma 5, since $\gamma_k \to 0$, there exists $j \geq 1$ such that

$$\rho(\Pi_n) \leq \rho(\Pi_j) \exp(-\lambda_m(\tau_n - \tau_j)).$$

Therefore, the initial error is bounded by

$$\rho(Y_n) \leq \rho(\Pi_j) \exp(\lambda_m \tau_j)\|\Theta_0\|_2 \exp(d_n) \quad \text{with} \quad d_n = -\lambda_m \tau_n - \log(\sqrt{\gamma_n}).$$

Using Lemma 6, we can treat the two cases $\beta < 1$ and $\beta = 1$. On the one hand, if $\beta < 1$ then we always have $d_n \to -\infty$. On the other hand, if $\beta = 1$, we have $d_n \sim \left(\frac{1}{2} - \gamma\lambda_m\right)\log(n)$ and the condition $2\alpha\lambda_m - 1 > 0$ ensures $d_n \to -\infty$. In both cases we get $\exp(d_n) \to 0$ and the initial error vanishes to 0.

**Weak convergence of the sampling error.**
Consider the random process

$$M_n = \gamma_n^{-1/2} \sum_{k=1}^n \gamma_k \Pi_{n,k} \sqrt{H} C_{k-1} w_k \mathbb{1}_{\mathcal{A}_{k-1}}.$$

Note that $\theta_k$, $\mathcal{A}_k$ and $C_k$ are $\mathcal{F}_k$-measurable. As a consequence, $M_n$ is a sum of martingale increments and we may rely on the following central limit theorem for martingale arrays.

**Theorem 4.** *(Hall & Heyde, 1980, Corollary 3.1) Let $(W_{n,i})_{1 \leq i \leq n, n \geq 1}$ be a triangular array of random vectors such that*

$$\mathbb{E}[W_{n,i} \mid \mathcal{F}_{i-1}] = 0, \quad \text{for all } 1 \leq i \leq n, \tag{6}$$

$$\sum_{i=1}^n \mathbb{E}[W_{n,i} W_{n,i}^\top \mid \mathcal{F}_{i-1}] \to V^* \geq 0, \quad \text{in probability}, \tag{7}$$

$$\sum_{i=1}^n \mathbb{E}[\|W_{n,i}\|^2 \mathbb{1}_{\{\|W_{n,i}\| > \varepsilon\}} \mid \mathcal{F}_{i-1}] \to 0, \quad \text{in probability}, \tag{8}$$

*then, $\sum_{i=1}^n W_{n,i} \rightsquigarrow \mathcal{N}(0, V^*)$, as $n \to \infty$.*

We start by verifying (7). Let $D_k = \sqrt{H} C_{k-1} \Gamma_{k-1} C_{k-1}^T \sqrt{H} \mathbb{1}_{\mathcal{A}_{k-1}} \in \mathcal{S}_d(\mathbb{R})$. The quadratic variation of $M_n$ is given by

$$\Sigma_n = \gamma_n^{-1} \sum_{k=1}^n \gamma_k^2 \Pi_{n,k} D_k \Pi_{n,k}^\top.$$

First we can check that $\Sigma_n$ is bounded. Using the triangle inequality and since the operator norm is submultiplicative, we have

$$\|\Sigma_n\| \leq \gamma_n^{-1} \sum_{k=1}^n \gamma_k^2 \|\Pi_{n,k} D_k \Pi_{n,k}^T\| \leq \gamma_n^{-1} \sum_{k=1}^n \gamma_k^2 \|D_k\| \|\Pi_{n,k}\|^2 = \gamma_n^{-1} \sum_{k=1}^n \gamma_k^2 \|D_k\| \rho(\Pi_{n,k})^2,$$

where we use in the last equality that $\Pi_{n,k}$ is real symmetric so $\|\Pi_{n,k}\| = \rho(\Pi_{n,k})$. On the event $\mathcal{A}_{k-1}$, the matrices $C_{k-1}$ and $\Gamma_{k-1}$ are bounded as $\|C_{k-1}\| \leq 2\|C\|$ and $\|\Gamma_{k-1}\| \leq 2\|\Gamma\|$ leading to the following bound for the matrix $D_k$,

$$\begin{aligned}
\|D_k\| &= \|\sqrt{H} C_{k-1} \Gamma_{k-1} C_{k-1}^T \sqrt{H} \mathbb{1}_{\mathcal{A}_{k-1}}\| \\
&\leq \|H\| \|\Gamma_{k-1}\| \|C_{k-1}\|^2 \mathbb{1}_{\mathcal{A}_{k-1}} \\
&\leq 8\|H\| \|\Gamma\| \|C\|^2 = U_D.
\end{aligned}$$

It follows that

$$\|\Sigma_n\| \le U_D \gamma_n^{-1} \sum_{k=1}^{n} \gamma_k^2 \rho(\Pi_{n,k})^2.$$

In view of Lemma 5, we shall split the summation from $k = 1, \dots, j$ and $k = j+1, \dots, n$ as

$$\gamma_n^{-1} \sum_{k=1}^{n} \gamma_k^2 \rho\left(\Pi_{n,k}\right)^2 = \gamma_n^{-1} \sum_{k=1}^{j} \gamma_k^2 \rho\left(\Pi_{n,k}\right)^2 + \gamma_n^{-1} \sum_{k=j+1}^{n} \gamma_k^2 \rho\left(\Pi_{n,k}\right)^2$$

$$\le \underbrace{\gamma_n^{-1} \sum_{k=1}^{j} \gamma_k^2 \rho\left(\Pi_{n,k}\right)^2}_{a_n} + \underbrace{\gamma_n^{-1} \sum_{k=j+1}^{n} \gamma_k^2 \prod_{i=k+1}^{n} \left(1 - \lambda_m \gamma_i\right)^2}_{b_n}.$$

For the first term $a_n$, we have for all $k = 1, \dots, j$

$$\rho(\Pi_{n,k}) \le \rho(\Pi_{n,j}) \le \prod_{i=j+1}^{n} \left(1 - \lambda_m \gamma_i\right) \le \exp(-\lambda_m(\tau_n - \tau_j)),$$

which implies since $(\gamma_k)$ is decreasing with $\gamma_1 = \alpha$ that

$$\sum_{k=1}^{j} \gamma_k^2 \rho\left(\Pi_{n,k}\right)^2 \le \alpha \tau_j \exp(-2\lambda_m(\tau_n - \tau_j)).$$

Therefore, similarly to the initial error term, we get

$$a_n \le \alpha \tau_j \exp(2\lambda_m \tau_j)) \exp(d_n) \quad \text{with} \quad d_n = -2\lambda_m \tau_n - \log(\gamma_n),$$

and the condition $2\alpha\lambda_m - 1 > 0$ ensures $d_n \to -\infty$ so that $a_n$ goes to 0 and is almost surely bounded by $U_a$.

For the second term $b_n$, we can apply Lemma 3 and need to distinguish between the two cases:

• ($\beta = 1$) If $\gamma_n = \alpha/n$, since $2\alpha\lambda_m > 1$, we can apply Lemma 3 ($p = 1, m = 2, \lambda = \lambda_m \alpha, x_j = 0, \varepsilon_k = \alpha^2$) and obtain

$$b_n \le \frac{\alpha^2}{2\alpha\lambda_m - 1} = U_b.$$

• ($\beta < 1$) If $\gamma_n = \gamma/n^\beta$, we deduce the same as before because $\lambda_m > 0$.

Finally in both cases, we get

$$\|\Sigma_n\| \le U_D \left(U_a + U_b\right). \tag{9}$$

We now derive the limit of $\Sigma_n$. We shall use a recursion equation to recover a stochastic approximation scheme. Note that

$$\gamma_n \Sigma_n = \sum_{k=1}^{n} \gamma_k^2 \Pi_{n,k} D_k \Pi_{n,k}^T \tag{10}$$

$$= \gamma_n^2 D_n + A_n \left(\sum_{k=1}^{n-1} \gamma_k^2 \Pi_{n-1,k} D_k \Pi_{n-1,k}^T\right) A_n^\top, \tag{11}$$

and recognize

$$\gamma_n \Sigma_n = \gamma_n^2 D_n + \gamma_{n-1} A_n \Sigma_{n-1} A_n^\top.$$

Replacing the symmetric matrix $A_n = I - \gamma_n \widetilde{K}$, we get (because $\Sigma_n$ is bounded almost surely)

$$\gamma_n \Sigma_n = \gamma_n^2 D_n + \gamma_{n-1}(I - \gamma_n \widetilde{K})\Sigma_{n-1}(I - \gamma_n \widetilde{K})$$
$$= \gamma_n^2 D_n + \gamma_{n-1}\left[\Sigma_{n-1} - \gamma_n \Sigma_{n-1}\widetilde{K} - \gamma_n \widetilde{K}\Sigma_{n-1} + O(\gamma_n^2)\right].$$

Divide by $\gamma_n$ to obtain

$$\Sigma_n = \gamma_n D_n + \frac{\gamma_{n-1}}{\gamma_n}\left[\Sigma_{n-1} - \gamma_n(\widetilde{K}\Sigma_{n-1} + \Sigma_{n-1}\widetilde{K}) + O(\gamma_n^2)\right],$$

and we recognize a stochastic approximation scheme

$$\Sigma_n = \Sigma_{n-1} - \gamma_n\left[\widetilde{K}\Sigma_{n-1} + \Sigma_{n-1}\widetilde{K} - D_n\right] + \frac{\gamma_{n-1} - \gamma_n}{\gamma_n}\Sigma_{n-1} + O(\gamma_{n-1}\gamma_n + |\gamma_{n-1} - \gamma_n|)$$

Recall that when $\beta < 1$ we have

$$\frac{1}{\gamma_n} - \frac{1}{\gamma_{n-1}} \to 0, \text{ i.e., } \frac{\gamma_{n-1} - \gamma_n}{\gamma_n} = o(\gamma_n).$$

- ($\beta = 1$) If $\gamma_n = \alpha/n$ we get

$$\Sigma_n = \Sigma_{n-1} - \frac{\alpha}{n}\left[\widetilde{K}\Sigma_{n-1} + \Sigma_{n-1}\widetilde{K} - \frac{1}{\alpha}\Sigma_{n-1} - D_n\right] + O(n^{-2})$$

$$\Sigma_n = \Sigma_{n-1} - \frac{\alpha}{n}\left[\left(\widetilde{K} - \frac{I}{2\alpha}\right)\Sigma_{n-1} + \Sigma_{n-1}\left(\widetilde{K} - \frac{I}{2\alpha}\right) - D_n\right] + O(n^{-2}).$$

- ($\beta < 1$) If $\gamma_n = \alpha/n^\beta$ we get

$$\Sigma_n = \Sigma_{n-1} - \gamma_n\left[\widetilde{K}\Sigma_{n-1} + \Sigma_{n-1}\widetilde{K} - D_n\right] + o(\gamma_n).$$

Recall that $\zeta = \mathbb{1}_{\{\beta=1\}}/(2\alpha)$ and define $\widetilde{K}_\zeta = \widetilde{K} - \zeta I$, so that in both cases, the recursion equation becomes

$$\Sigma_n = \Sigma_{n-1} - \gamma_n\left[\widetilde{K}_\zeta\Sigma_{n-1} + \Sigma_{n-1}\widetilde{K}_\zeta^\top - D_n\right] + o(\gamma_n).$$

We can vectorize this equation. The vectorization of an $m \times n$ matrix $A = (a_{i,j})$, denoted $\mathrm{vec}(A)$, is the $mn \times 1$ column vector obtained by stacking the columns of the matrix A on top of one another:

$$\mathrm{vec}(A) = [a_{1,1}, \dots, a_{m,1}, a_{1,2}, \dots, a_{m,2}, \dots, a_{1,n}, \dots, a_{m,n}]^T.$$

Applying this operator to our stochastic approximation scheme gives

$$\mathrm{vec}(\Sigma_n) = \mathrm{vec}(\Sigma_{n-1}) - \gamma_n\left[\mathrm{vec}\left(\widetilde{K}_\zeta\Sigma_{n-1} + \Sigma_{n-1}\widetilde{K}_\zeta^\top\right) - \mathrm{vec}(D_n)\right] + o(\gamma_n).$$

Denote by $\otimes$ the Kronecker product, we have the following property

$$\mathrm{vec}\left(K_\zeta\Sigma_{n-1} + \Sigma_{n-1}K_\zeta^\top\right) = \left(I_d \otimes K_\zeta + K_\zeta^\top \otimes I_d\right)\mathrm{vec}(\Sigma_{n-1}).$$

Define $D$ as the almost sure limit of $D_n$, *i.e.*

$$D = \lim_{n\to\infty} D_n = \sqrt{H}C\Gamma C\sqrt{H}.$$

Introduce $v_n = \mathrm{vec}(\Sigma_n)$ and $Q = \left(I_d \otimes \widetilde{K}_\zeta + \widetilde{K}_\zeta \otimes I_d\right)$. We have almost surely

$$v_n = v_{n-1} - \gamma_n\left(Qv_{n-1} - \mathrm{vec}(D)\right) + \gamma_n\mathrm{vec}(D_n - D) + o(\gamma_n)$$
$$= v_{n-1} - \gamma_n\left(Qv_{n-1} - \mathrm{vec}(D)\right) + \varepsilon_n\gamma_n$$

where $\varepsilon_n \to 0$ almost surely. This is a stochastic approximation scheme with the affine function $h(v) = Qv - \text{vec}(D)$ for $v \in \mathbb{R}^{d^2}$. Let $v^\star$ be the solution of $h(v) = 0$ which is well defined since $Q = \left( I_d \otimes \widetilde{K}_\zeta + \widetilde{K}_\zeta^\top \otimes I_d \right)$ is invertible. Indeed, the eigenvalues of $Q$ are $\mu_i + \mu_j$, $1 \leq i, j \leq d$, where the $\mu_i$, $i = 1, \ldots, d$ are the eigenvalues of $\widetilde{K}_\zeta$. Equivalently, the eigenvalues of $Q$ are of the form $(\lambda_i - \zeta) + (\lambda_j - \zeta)$ where the $\lambda_i$, $i = 1, \ldots, d$ are the eigenvalues of $\widetilde{K}$. Because $\lambda_m > \zeta$, we have that $Q \succ 0$. As a consequence

$$
\begin{aligned}
(v_n - v^\star) &= (v_{n-1} - v^\star) - \gamma_n \left( h(v_{n-1}) - h(v^\star) \right) + \varepsilon_n \gamma_n \\
&= (v_{n-1} - v^\star) - \gamma_n Q \left( v_{n-1} - v^\star \right) + \varepsilon_n \gamma_n \\
&= B_n \left( v_{n-1} - v^\star \right) + \varepsilon_n \gamma_n,
\end{aligned}
$$

with $B_n = (I_{d^2} - \gamma_n Q)$. By induction, we obtain

$$
(v_n - v^\star) = (B_n \ldots B_1)(v_0 - v^\star) + \sum_{k=1}^n \gamma_k (B_n \ldots B_{k+1}) \varepsilon_k,
$$

Define $\lambda_Q = \lambda_{\min}(Q) > 0$ and remark that

$$
\|B_n \ldots B_{k+1}\| \leq \prod_{j=k+1}^n \|B_j\| = \prod_{j=k+1}^n (1 - \gamma_j \lambda_Q).
$$

It follows that

$$
\begin{aligned}
\|v_n - v^\star\|_2 &\leq \|B_n \ldots B_1\| \|v_0 - v^\star\|_2 + \sum_{k=1}^n \gamma_k \|B_n \ldots B_{k+1}\| \|\varepsilon_k\|_2 \\
&\leq \prod_{j=1}^n (1 - \gamma_j \lambda_Q) \|v_0 - v^\star\|_2 + \sum_{k=1}^n \gamma_k \prod_{j=k+1}^n (1 - \gamma_j \lambda_Q) \|\varepsilon_k\|_2
\end{aligned}
$$

Applying Lemma 3 we obtain that the right-hand side term goes to 0. The left-hand side term goes to 0 under the effect of the product by definition of $(\gamma_k)_{k \geq 1}$. We therefore conclude that $v_n \to v^\star$ almost surely. From easy manipulation involving $\text{vec}(\cdot)$ and $\otimes$, this is equivalent to $\Sigma_n \to \Sigma$, where $\Sigma$ is the solution of the Lyapunov equation

$$
(\widetilde{K} - \zeta I)\Sigma + \Sigma(\widetilde{K} - \zeta I) = D.
$$

Now we turn our attention to (8). We need to show that almost surely,

$$
\gamma_n^{-1} \sum_{k=1}^n \gamma_k^2 \mathbb{E}[\|\Pi_{n,k} \sqrt{H} C_{k-1} w_k\|_2^2 \mathbb{1}_{\{\gamma_k \|\Pi_{n,k}\sqrt{H} C_{k-1} w_k\|_2 > \varepsilon \gamma_n^{1/2}\}} \mid \mathcal{F}_{k-1}] \mathbb{1}_{\mathcal{A}_{k-1}} \to 0.
$$

We have

$$
\begin{aligned}
&\mathbb{E}[\gamma_n^{-1} \gamma_k^2 \|\Pi_{n,k} \sqrt{H} C_{k-1} w_k\|_2^2 \mathbb{1}_{\{\gamma_k \|\Pi_{n,k}\sqrt{H} C_{k-1} w_k\|_2 > \varepsilon \gamma_n^{1/2}\}} \mid \mathcal{F}_{k-1}] \\
&\leq \varepsilon^{-\delta} \mathbb{E}[(\gamma_n^{-1/2} \gamma_k \|\Pi_{n,k} \sqrt{H} C_{k-1} w_k\|_2)^{2+\delta} \mid \mathcal{F}_{k-1}] \\
&\leq \varepsilon^{-\delta} (\gamma_n^{-1/2} \gamma_k \|\Pi_{n,k} \sqrt{H} C_{k-1}\|^{2+\delta} \mathbb{E}[\|w_k\|_2^{2+\delta} \mid \mathcal{F}_{k-1}].
\end{aligned}
$$

Let $U(\omega) = \sup_{k \geq 1} \mathbb{E}[\|w_k\|_2^{2+\delta} \mid \mathcal{F}_{k-1}] \mathbb{1}_{\mathcal{A}_{k-1}}$ which is almost surely finite by Assumption 5. We get

$$
\begin{aligned}
&\mathbb{E}[\gamma_n^{-1} \gamma_k^2 \|\Pi_{n,k} \sqrt{H} C_{k-1} w_k\|_2^2 \mathbb{1}_{\{\gamma_k \|\Pi_{n,k}\sqrt{H} C_{k-1} w_k\|_2 > \varepsilon \gamma_n^{1/2}\}} \mid \mathcal{F}_{k-1}] \mathbb{1}_{\mathcal{A}_{k-1}} \\
&\leq \varepsilon^{-\delta} \left( 2\|\sqrt{H}\| \|C\| \right)^{2+\delta} U(\omega) (\gamma_n^{-1/2} \gamma_k \rho(\Pi_{n,k}))^{2+\delta}
\end{aligned}
$$

Hence by showing that

$$\sum_{k=1}^{n} (\gamma_n^{-1/2} \gamma_k \rho(\Pi_{n,k}))^{2+\delta} \to 0,$$

we will obtain (8). The previous convergence can be deduced from Lemma 3 with $p = 1 + \delta/2$, $m = 2 + \delta$, $\epsilon_k = \gamma_k^{\delta/2}$, checking that $(2 + \delta)\alpha\lambda_m > 1 + \delta/2$.

**Step 2. Proof of Equation** (5).

A preliminary step to the derivation of Equation (5) is to obtain that $\widetilde{\Delta}_k \to 0$ almost surely. For any $\theta$ and $\eta$ in $\mathbb{R}^d$, we have

$$\|\theta\|^2 = \|\eta\|^2 + 2\eta^\top(\theta - \eta) + \|\theta - \eta\|^2$$

implying that for all $k \geq 0$

$$\mathbb{E}[\|\Theta_{k+1}\|^2 | \mathcal{F}_k] = \|\widetilde{\Theta}_k\|^2 - 2\gamma_{k+1}\Theta_k^\top \widetilde{K}\Theta_k + \gamma_{k+1}^2 \mathbb{E}[\|\widetilde{K}\Theta_k - C_k w_{k+1}\mathbb{1}_{\mathcal{A}_k}\|^2 | \mathcal{F}_k].$$

Since $(w_k)$ is a martingale increment and because on $\mathcal{A}_k$, $\rho(C_k) \leq 2\rho(C)$, we get

$$\begin{aligned}
\mathbb{E}[\|\widetilde{K}\Theta_k - C_k w_{k+1}\mathbb{1}_{\mathcal{A}_k}\|^2 | \mathcal{F}_k] &= \mathbb{E}[\|\widetilde{K}\Theta_k\|^2 | \mathcal{F}_k] + \mathbb{E}[\|C_k w_{k+1}\mathbb{1}_{\mathcal{A}_k}\|^2 | \mathcal{F}_k] \\
&\leq \lambda_M^2 \|\Theta_k\|^2 + \rho(C_k)^2 \mathbb{E}[\|w_{k+1}\mathbb{1}_{\mathcal{A}_k}\|^2 | \mathcal{F}_k] \\
&\leq \lambda_M^2 \|\Theta_k\|^2 + 4\rho(C)^2 \mathbb{E}[\|w_{k+1}\mathbb{1}_{\mathcal{A}_k}\|^2 | \mathcal{F}_k],
\end{aligned}$$

Injecting this bound in the previous equality yields

$$\mathbb{E}[\|\Theta_{k+1}\|^2 | \mathcal{F}_k] \leq \|\Theta_k\|^2(1 + \gamma_{k+1}^2 \lambda_M^2) - 2\gamma_{k+1}\Theta_k^\top \widetilde{K}\Theta_k + 4\rho(C)^2\gamma_{k+1}^2 \mathbb{E}[\|w_{k+1}\|^2 | \mathcal{F}_k]\mathbb{1}_{\mathcal{A}_k}.$$

Since, using (3),

$$\sum_{k \geq 0} \gamma_{k+1}^2 \mathbb{E}[\|w_{k+1}\|^2 | \mathcal{F}_k]\mathbb{1}_{\mathcal{A}_k} \leq \left( \sup_{k \geq 0} \mathbb{E}[\|w_{k+1}\|^2 | \mathcal{F}_k]\mathbb{1}_{\mathcal{A}_k} \right) \left( \sum_{k \geq 0} \gamma_{k+1}^2 \right) < \infty,$$

we are in position to apply the Robbins-Siegmund Theorem 6 and we obtain the almost sure convergence of $\sum_k \gamma_{k+1}\Theta_k^\top \widetilde{K}\Theta_k$ and $\|\Theta_k\|_2^2 \to V_\infty$. Because $\widetilde{K}$ is positive definite, it gives that, with probability 1, $\sum_{k \geq 0} \gamma_{k+1}\|\Theta_k\|^2 < +\infty$, from which, we deduce $\liminf_k \|\Theta_k\|^2 = 0$. Therefore one can extract a subsequence $\Theta_k$ such that $\|\Theta_k\|^2 \to 0$. Using the above second condition yields $V_\infty = 0$ and we conclude that $\widetilde{\Delta}_k = H^{-1/2}\Theta_k \to 0$.

Define the difference

$$E_k = \Delta_k - \widetilde{\Delta}_k.$$

Since $\theta \mapsto \nabla^2 F(\theta)$ is continous at $\theta^\star$, we can apply a coordinate-wise mean value theorem. Indeed, for any $\theta \in \mathbb{R}^d$, we have $\nabla F(\theta) = (\partial_1 F(\theta), \ldots, \partial_d F(\theta))$ where for all $j = 1, \ldots, d$, the partial derivatives functions $\partial_j F : \mathbb{R}^d \to \mathbb{R}$ are Lipschitz continuous. Denote by $\nabla(\partial_j F) : \mathbb{R}^d \to \mathbb{R}^d$ the gradient of the partial derivative $\partial_j F$, i.e., $\nabla(\partial_j F)(\theta) = (\partial_{1,j}^2 F(\theta), \ldots, \partial_{d,j}^2 F(\theta))$. For any $\theta, \eta \in B(\theta^\star, \varepsilon)$, there exists $\xi_j \in \mathbb{R}^d$ such that

$$\partial_j F(\theta) - \partial_j F(\eta) = \nabla(\partial_j F)(\xi_j)(\theta - \eta).$$

We construct a Hessian matrix by rows $H(\xi) = H(\xi_1, \ldots, \xi_d)$ where the $j$-th row is equal to $\nabla(\partial_j F)(\xi_j) = (\partial_{1,j}^2 F(\xi_j), \ldots, \partial_{d,j}^2 F(\xi_j))$

$$H(\xi) = \begin{bmatrix} \partial_{1,1}^2 F(\xi_1) & \cdots & \partial_{1,d}^2 F(\xi_1) \\ \vdots & \ddots & \vdots \\ \partial_{d,1}^2 F(\xi_d) & \cdots & \partial_{d,d}^2 F(\xi_d) \end{bmatrix}$$

and we can write

$$\nabla F(\theta) - \nabla F(\eta) = H(\xi)(\theta - \eta).$$

There exists $\xi_k = (\xi_k^{(1)}, \ldots, \xi_k^{(d)})$ with $\xi_k^{(j)} \in [\theta^\star + E_k, \theta_k]$ and $\xi_k' = (\xi_k'^{(1)}, \ldots, \xi_k'^{(d)})$ with $\xi_k'^{(j)} \in [\theta^\star + E_k, \theta^\star]$ such that

$$\nabla F(\theta^\star + E_k) - \nabla F(\theta_k) = -H(\xi_k)\widetilde{\Delta}_k \tag{12}$$
$$\nabla F(\theta^\star + E_k) = H(\xi_k')E_k. \tag{13}$$

Let $\eta > 0$ such that $2\alpha\lambda_m(1 - 3\eta) > 1$. This choice will come clear at the end of the reasoning. On the one hand, we have $C_k \to C$. On the other hand, using Lemma 4, the spectrum of $C_k H$ is real and positive. Hence, we have the convergence of the eigenvalues of $C_k H$ towards the eigenvalues of $K = CH$. This follows from the definition of eigenvalues as roots of the characteristic polynomial and the fact that the roots of any polynomial $P \in \mathbb{C}[X]$ are continuous functions of the coefficients (Zedek, 1965). Consequently, there exists $n_1(\omega)$ such that for all $k \geq n_1(\omega)$,

$$(1 - \eta)\lambda_m \leq \lambda_{\min}(C_k H) \leq \lambda_{\max}(C_k H) \leq (1 + \eta)\lambda_M. \tag{14}$$

We can define $n_2(\omega)$ such that for all $k \geq n_2(\omega)$

$$\mathcal{A}_k \text{ is realized.} \tag{15}$$

Since $\|\sqrt{H^{-1}}H(\xi_k')\sqrt{H^{-1}} - I_d\| \to 0$ as $k \to \infty$, there is $n_3(\omega)$ and $n_4(\omega)$ such that for all $k \geq n_3(\omega)$

$$\|\sqrt{H^{-1}}H(\xi_k')\sqrt{H^{-1}} - I_d\| \leq \frac{\eta}{1+\eta}\frac{\lambda_m}{\lambda_M}, \tag{16}$$

and for all $k \geq n_4(\omega)$,

$$\|\sqrt{H^{-1}}H(\xi_k')\sqrt{H^{-1}}\| \leq 1. \tag{17}$$

Since $\gamma_k \to 0$, there is $n_5$ such that for all $k \geq n_5$

$$\gamma_{k+1} \leq \frac{2\eta\lambda_m}{(1+\eta)^2\lambda_M^2}. \tag{18}$$

To use the previous local properties, define $n_0(\omega) = n_1(\omega) \vee n_2(\omega) \vee n_3(\omega) \vee n_4(\omega) \vee n_5$ and introduce the set $\mathcal{E}_j$ along with its complement $\mathcal{E}_j^c$, defined by

$$\mathcal{E}_j = \{\omega : j \geq n_0(\omega)\}.$$

Let $\delta > 0$ and take $j \geq 1$ large enough such that $\mathbb{P}(\mathcal{E}_j^c) \leq \delta$. Invoking the Markov inequality, we have for all $a > 0$

$$\mathbb{P}(\gamma_k^{-1/2}\|E_k\| > a) = \mathbb{P}(\gamma_k^{-1/2}\|E_k\| > a, \mathcal{E}_j) + \mathbb{P}(\gamma_k^{-1/2}\|E_k\| > a, \mathcal{E}_j^c)$$
$$\leq \mathbb{P}(\gamma_k^{-1/2}\|E_k\| > a, \mathcal{E}_j) + \delta$$
$$\leq \gamma_k^{-1/2}a^{-1}\mathbb{E}[\|E_k\|\mathbb{1}_{\mathcal{E}_j}] + \delta$$

Because $\delta$ is arbitrary, we only need to show that for any value of $j \geq 1$,

$$e_k := \mathbb{E}[\|E_k\|\mathbb{1}_{\mathcal{E}_j}] = o(\gamma_k^{1/2}).$$

To prove this fact, we shall recognize a stochastic algorithm for the sequence $e_k$.

Let $k \geq j$ and assume further that $\mathcal{E}_j$ is realized. We have, because of (15),

$$E_{k+1} = \Delta_k - \widetilde{\Delta}_k - \gamma_{k+1}C_k\nabla F(\theta_k) + \gamma_{k+1}K\widetilde{\Delta}_k.$$

Introducing $\widetilde{E}_k = \sqrt{H}E_k$, we find

$$\widetilde{E}_{k+1} = \widetilde{E}_k - \gamma_{k+1}\sqrt{H}C_k\nabla F(\theta_k) + \gamma_{k+1}\sqrt{H}K\widetilde{\Delta}_k,$$

and using (12), it comes that

$$\widetilde{E}_{k+1} = \widetilde{E}_k - \gamma_{k+1}\sqrt{H}C_k\nabla F(\theta^\star + E_k) - \gamma_{k+1}\sqrt{H}C_kH(\xi_k)\widetilde{\Delta}_k + \gamma_{k+1}\sqrt{H}K\widetilde{\Delta}_k$$
$$= \widetilde{E}_k - \gamma_{k+1}\sqrt{H}C_k\nabla F(\theta^\star + E_k) + \gamma_{k+1}\sqrt{H}(K - C_kH(\xi_k))\widetilde{\Delta}_k.$$

Using Minkowski inequality, we have

$$\|\widetilde{E}_{k+1}\| \leq \|\widetilde{E}_k - \gamma_{k+1}\sqrt{H}C_k\nabla F(\theta^\star + E_k)\| + \|\gamma_{k+1}\sqrt{H}(K - C_kH(\xi_k))\widetilde{\Delta}_k\|.$$

We shall now focus on the first term. Still on the set $\mathcal{E}_j$, we have

$$\|\widetilde{E}_k - \gamma_{k+1}\sqrt{H}C_k\nabla F(\theta^\star + E_k)\|^2$$
$$= \|\widetilde{E}_k\|^2 - 2\gamma_{k+1}\langle\widetilde{E}_k, \sqrt{H}C_k\nabla F(\theta^\star + E_k)\rangle + \gamma_{k+1}^2\|\sqrt{H}C_k\nabla F(\theta^\star + E_k)\|^2 \qquad (19)$$

We have on the one hand using (13)

$$\langle\widetilde{E}_k, \sqrt{H}C_k\nabla F(\theta^\star + E_k)\rangle = \langle\widetilde{E}_k, \sqrt{H}C_kH(\xi_k')E_k\rangle$$
$$= \langle\widetilde{E}_k, \sqrt{H}C_kHE_k\rangle + \langle\widetilde{E}_k, \sqrt{H}C_k(H(\xi_k') - H)E_k\rangle$$

Due to (14), the first term satisfies

$$\langle\widetilde{E}_k, \sqrt{H}C_kHE_k\rangle = \langle\widetilde{E}_k, \sqrt{H}C_k\sqrt{H}\widetilde{E}_k\rangle$$
$$\geq \lambda_{\min}(C_kH)\|\widetilde{E}_k\|^2$$
$$\geq (1 - \eta)\lambda_m\|\widetilde{E}_k\|^2$$

The second term satisfies

$$\langle\widetilde{E}_k, \sqrt{H}C_k(H(\xi_k') - H)E_k\rangle = \langle\widetilde{E}_k, \sqrt{H}C_k\sqrt{H}(\sqrt{H^{-1}}H(\xi_k')\sqrt{H^{-1}} - I_d)\widetilde{E}_k\rangle$$
$$\geq -\left|\langle\widetilde{E}_k, \sqrt{H}C_k\sqrt{H}(\sqrt{H^{-1}}H(\xi_k')\sqrt{H^{-1}} - I_d)\widetilde{E}_k\rangle\right|$$

Using Cauchy-Schwarz inequality, the submultiplicativity of the norm, (14) and (16), we have

$$\left|\langle\widetilde{E}_k, \sqrt{H}C_k\sqrt{H}(\sqrt{H^{-1}}H(\xi_k')\sqrt{H^{-1}} - I_d)\widetilde{E}_k\rangle\right|$$
$$\leq \|\sqrt{H}C_k\sqrt{H}\|\|\sqrt{H^{-1}}H(\xi_k')\sqrt{H^{-1}} - I_d\|\|\widetilde{E}_k\|^2$$
$$\leq \eta\lambda_m\|\widetilde{E}_k\|^2.$$

Finally, it follows that

$$\langle\widetilde{E}_k, \sqrt{H}C_k\nabla F(\theta^\star + E_k)\rangle \geq (1 - 2\eta)\lambda_m\|\widetilde{E}_k\|^2 \qquad (20)$$

On the other hand using (13), (14) and (17),

$$\|\sqrt{H}C_k\nabla F(\theta^\star + E_k)\|^2 = \|\sqrt{H}C_kH(\xi_k')E_k\|^2$$
$$= \|\sqrt{H}C_k\sqrt{H}(\sqrt{H^{-1}}H(\xi_k')\sqrt{H^{-1}})\widetilde{E}_k\|^2$$
$$\leq \lambda_{\max}(C_kH)^2\|\sqrt{H^{-1}}H(\xi_k')\sqrt{H^{-1}}\|^2\|\widetilde{E}_k\|^2$$
$$\leq (1 + \eta)^2\lambda_M^2\|\widetilde{E}_k\|^2 \qquad (21)$$

Putting together (19), (20), (21) and using (18) gives that, on $\mathcal{E}_j$,

$$\|\widetilde{E}_k - \gamma_{k+1}\sqrt{H}C_k\nabla F(\theta^\star + E_k)\|^2$$
$$\leq \|\widetilde{E}_k\|^2(1 - 2\gamma_{k+1}(1-2\eta)\lambda_m + \gamma_{k+1}^2(1+\eta)^2\lambda_M^2)$$
$$\leq \|\widetilde{E}_k\|^2(1 - 2\gamma_{k+1}(1-3\eta)\lambda_m).$$

By the Minkowski inequality and the fact that $(1-x)^{1/2} \leq 1 - x/2$, on $\mathcal{E}_j$, it holds

$$\|\widetilde{E}_{k+1}\| \leq \|\widetilde{E}_k\|(1 - 2\gamma_{k+1}(1-3\eta)\lambda_m)^{1/2} + \gamma_{k+1}\|\sqrt{H}(K - C_kH(\xi_k))\widetilde{\Delta}_k\|$$
$$\leq \|\widetilde{E}_k\|(1 - \gamma_{k+1}(1-3\eta)\lambda_m) + \gamma_{k+1}\|\sqrt{H}\|\|(K - C_kH(\xi_k))\widetilde{\Delta}_k\|$$

Hence, we have shown that for any $k \geq j$,

$$\|\widetilde{E}_k\|\mathbb{1}_{\mathcal{E}_j} \leq \|\widetilde{E}_k\|\mathbb{1}_{\mathcal{E}_j}(1 - \gamma_{k+1}(1-3\eta)\lambda_m) + \gamma_{k+1}\|\sqrt{H}\|\|(K - C_kH(\xi_k))\mathbb{1}_{\mathcal{E}_j}\widetilde{\Delta}_k\|.$$

It follows that, for any $k \geq j$,

$$e_{k+1} \leq e_k(1 - \gamma_{k+1}(1-3\eta)\lambda_m) + \gamma_{k+1}\|\sqrt{H}\|\mathbb{E}[\|U_k\widetilde{\Delta}_k\|],$$

with $U_k = (K - C_kH(\xi_k))\mathbb{1}_{\mathcal{E}_j}$. Because with probability 1, $\|U_k\|$ is bounded, we can apply the Lebesgue dominated convergence theorem to obtain that $\varepsilon_k = \mathbb{E}[\|U_k\|^2] \to 0$. From the Cauchy-Schwarz inequality, we get

$$\mathbb{E}[\|U_k\widetilde{\Delta}_k\|] \leq \sqrt{\varepsilon_k}\sqrt{\mathbb{E}[\|\widetilde{\Delta}_k\|_2^2]}.$$

On the other hand, we have already shown in (9) that $\rho(\Sigma_k) = \|\Sigma_k\| \leq U_D(U_a + U_b)$. Since $\widetilde{\Delta}_k = \sqrt{H^{-1}}\Theta_k = \sqrt{H^{-1}}\sqrt{\gamma_k}(Y_k + M_k)$, we have

$$\mathbb{E}[\|\widetilde{\Delta}_k\|_2^2] \leq 2(\gamma_k/\lambda_m)(\|Y_k\|_2^2 + \mathbb{E}[\|M_k\|_2^2]),$$

where the last term is the leading term and satisfies

$$\mathbb{E}[\|M_k\|_2^2]] = \mathbb{E}[\text{Tr}(\Sigma_k)] \leq d\mathbb{E}[\rho(\Sigma_k)].$$

Therefore, we have

$$\mathbb{E}[\|\widetilde{\Delta}_k\|_2^2] \leq \gamma_k A$$

for some $A > 0$. Consequently, for all $k \geq j$,

$$e_{k+1} \leq e_k(1 - \gamma_{k+1}(1-3\eta)\lambda_m) + \gamma_{k+1}^{3/2}A'\|\sqrt{H}\|\varepsilon_k^{1/2}.$$

The condition $2\alpha\lambda_m(1-3\eta) > 1$ ensures that we can apply Lemma 3 with $(m\lambda > p), m = 1, p = 1/2, \lambda = \alpha(1-3\eta)\lambda_m$. we finally get

$$\limsup_k(e_k/\gamma_k^{1/2}) = 0.$$

As a consequence, $e_k = o(\sqrt{\gamma_k})$, which concludes the proof.

Since $\gamma_k^{-1/2}\sqrt{H}\widetilde{\Delta}_k \to \mathcal{N}(0, \Sigma)$, we have $\gamma_k^{-1/2}\widetilde{\Delta}_k \to \mathcal{N}(0, \widetilde{\Sigma})$ where $\widetilde{\Sigma} = \sqrt{H^{-1}}\Sigma\sqrt{H^{-1}}$. Recall that $\Sigma$ satisfies the Lyapunov equation

$$(\sqrt{H}C\sqrt{H} - \varsigma I_d)\Sigma + \Sigma(\sqrt{H}C\sqrt{H} - \varsigma I_d) = \sqrt{H}C\Gamma C\sqrt{H}.$$

Multiplying on the left and right sides by $\sqrt{H^{-1}}$, we get

$$C\sqrt{H}\Sigma\sqrt{H^{-1}} - \varsigma\sqrt{H^{-1}}\Sigma\sqrt{H^{-1}} + \sqrt{H^{-1}}\Sigma\sqrt{H}C - \varsigma\sqrt{H^{-1}}\Sigma\sqrt{H^{-1}} = C\Gamma C,$$

where we recognize the following Lyapunov equation

$$(CH - \varsigma I_d)\widetilde{\Sigma} + \widetilde{\Sigma}(CH - \varsigma I_d)^\top = C\Gamma C.$$

## A.2 Proof of the almost sure convergence (Theorem 3)

The idea behind the proof of the almost sure convergence is to apply the Robbins-Siegmund Theorem (Theorem 6) (which can be found in Appendix C) in combination with the following key deterministic result.

**Lemma 1** (Deterministic result). *Let $F : \mathbb{R}^d \to \mathbb{R}$ be a $L$-smooth function and $(\theta_t)$ a random sequence obtained by the SGD update rule $\theta_{t+1} = \theta_t - \gamma_{t+1}C_t g_t$ where $(\gamma)_{t \geq 1}$ a positive sequence of learning rates and $C_{t-1} \preceq \nu_t I_d$ are such that $\sum_t \gamma_t \nu_t = \infty$. Let $\omega \in \Omega$ such that the following limits exist:*

$$(i) \ \sum_{t \geq 0} \gamma_{t+1}\nu_{t+1}\|\nabla F(\theta_t(\omega))\|_2^2 < \infty \quad (ii) \ \sum_{t \geq 1} \gamma_t C_{t-1}(g_{t-1}(\omega) - \nabla F(\theta_{t-1}(\omega))) < \infty$$

*then $\nabla F(\theta_t(\omega)) \to 0$ as $t \to \infty$.*

**Proof.** The proof (and in particular the reasoning by contradiction) is inspired from the proof of Proposition 1 in Bertsekas & Tsitsiklis (2000). For ease of notation we omit the $\omega$ in the proof. Note that condition (i) along with $\sum_t \gamma_t \nu_t = \infty$ implie that $\liminf_t \|\nabla F(\theta_t)\| = 0$. Now, by contradiction, let $\varepsilon > 0$ and assume that

$$\limsup_t \|\nabla F(\theta_t)\| > \varepsilon$$

We have that there is infinitely many $t$ such that $\|\nabla F(\theta_t)\| < \varepsilon/2$ and also infinitely many $t$ such that $\|\nabla F(\theta_t)\| > \varepsilon$. It follows that there is infinitely many *crossings* between the sets $\{t \in \mathbb{N} : \|\nabla F(\theta_t)\| < \varepsilon/2\}$ and $\{t \in \mathbb{N} : \|\nabla F(\theta_t)\| > \varepsilon\}$. A *crossing* is a collection of indexes $I_k = \{L_k, L_k+1, \ldots, U_k-1\}$ with $L_k \leq U_k$ ($I_k = \emptyset$ when $L_k = U_k$) such that for all $t \in I_k$,

$$\|\nabla F(\theta_{L_k-1})\| < \varepsilon/2 \leq \|\nabla F(\theta_t)\| \leq \varepsilon < \|\nabla F(\theta_{U_k})\|.$$

Define the following partial Cauchy sequence $R_k = \sum_{t=L_k}^{U_k} \gamma_t(g_{t-1} - \nabla F(\theta_{t-1}))$ and note that condition (ii) implies that $R_k \to 0$ as $k \to \infty$. For all $k \geq 1$,

$$\begin{aligned}
\varepsilon/2 &\leq \|\nabla F(\theta_{U_k})\|_2 - \|\nabla F(\theta_{L_k-1})\|_2 \\
&\leq \|\nabla F(\theta_{U_k}) - \nabla F(\theta_{L_k-1})\|_2 \\
&\leq L\|\theta_{U_k} - \theta_{L_k-1}\|_2,
\end{aligned}$$

where we use that $\nabla F$ is $L$-Lipschitz. Then using the update rule $\theta_t - \theta_{t-1} = -\gamma_t C_{t-1} g_{t-1}$, we have by sum

$$\begin{aligned}
\varepsilon/2 \leq L\|\sum_{t=L_k}^{U_k} \theta_t - \theta_{t-1}\|_2 &= L\|\sum_{t=L_k}^{U_k} \gamma_t C_{t-1} g_{t-1}\|_2 \\
&\leq L\|\sum_{t=L_k}^{U_k} \gamma_t C_{t-1}\nabla F(\theta_{t-1})\|_2 + L\|\sum_{t=L_k}^{U_k} \gamma_t C_{t-1}(g_{t-1} - \nabla F(\theta_{t-1}))\|_2 \\
&\leq L \sum_{t=L_k}^{U_k} \gamma_t \nu_t \|\nabla F(\theta_{t-1})\|_2 + L\|R_k\|_2
\end{aligned}$$

Since in the previous equation $\|\nabla F(\theta_{t-1})\|_2 > \varepsilon/2$, we get

$$(\varepsilon/2)^2 \leq L \sum_{t=L_k}^{U_k} \gamma_t \nu_t \|\nabla F(\theta_{t-1})\|_2^2 + (\varepsilon/2)L\|R_k\|_2$$

But since $\sum_{t \geq 0} \gamma_{t+1}\nu_{t+1}\|\nabla F(\theta_t)\|^2$ is finite and $\lim_k R_k = 0$, the previous upper bound goes to 0 and implies a contradiction. $\square$

It remains to show that points (i) and (ii) in Lemma 1 are valid with probability one. Since $\theta \mapsto F(\theta)$ is $L$-smooth, we have the quadratic bound (see Nesterov (2013))

$$\forall \theta, \eta \in \mathbb{R}^d \quad F(\eta) \leq F(\theta) + \langle \nabla F(\theta), \eta - \theta \rangle + \frac{L}{2} \|\eta - \theta\|_2^2.$$

Using the update rule $\theta_{k+1} = \theta_k - \gamma_{k+1} C_k g(\theta_k, \xi_{k+1})$, we get

$$F(\theta_{k+1}) \leq F(\theta_k) + \langle \nabla F(\theta_k), \theta_{k+1} - \theta_k \rangle + \frac{L}{2} \|\theta_{k+1} - \theta_k\|_2^2$$

$$= F(\theta_k) - \gamma_{k+1} \langle \nabla F(\theta_k), C_k g(\theta_k, \xi_{k+1}) \rangle + \frac{L}{2} \gamma_{k+1}^2 \|C_k g(\theta_k, \xi_{k+1})\|_2^2.$$

The last term can be upper bounded using the matrix norm and Assumption 9 as

$$\|C_k g(\theta_k, \xi_{k+1})\|_2^2 \leq \|C_k\|^2 \|g(\theta_k, \xi_{k+1})\|_2^2 \leq \nu_{k+1}^2 \|g(\theta_k, \xi_{k+1})\|_2^2,$$

and we have the inequality

$$F(\theta_{k+1}) \leq F(\theta_k) - \gamma_{k+1} \langle \nabla F(\theta_k), C_k g(\theta_k, \xi_{k+1}) \rangle + \frac{L}{2} (\gamma_{k+1} \nu_{k+1})^2 \|g(\theta_k, \xi_{k+1})\|_2^2.$$

Introduce $u_k = \gamma_k \nu_k$ and $v_k = \gamma_k \mu_k$, we have $\sum_{k \geq 1} v_k = +\infty$ and $\sum_{k \geq 1} u_k^2 < +\infty$ a.s. in virtue of Assumption 9. The random variables $F(\theta_k), C_k$ are $\mathcal{F}_k$-measurable and the gradient estimate is unbiased with respect to $\mathcal{F}_k$. Taking the conditional expectation denoted by $\mathbb{E}_k$ leads to

$$\mathbb{E}_k \left[ F(\theta_{k+1}) \right] - F(\theta_k) \leq -\gamma_{k+1} \langle \nabla F(\theta_k), \mathbb{E}_k \left[ C_k g(\theta_k, \xi_{k+1}) \right] \rangle + \frac{L}{2} u_{k+1}^2 \mathbb{E}_k \left[ \|g(\theta_k, \xi_{k+1})\|_2^2 \right]$$

$$= -\gamma_{k+1} \nabla F(\theta_k)^\top C_k \nabla F(\theta_k) + \frac{L}{2} u_{k+1}^2 \mathbb{E}_k \left[ \|g(\theta_k, \xi_{k+1})\|_2^2 \right].$$

On the one hand for the first term, using Assumption 9 ,

$$\nabla F(\theta_k)^\top C_k \nabla F(\theta_k) \geq \lambda_{\min}(C_k) \|\nabla F(\theta_k)\|_2^2 \geq \mu_{k+1} \|\nabla F(\theta_k)\|_2^2.$$

On the other hand, using Assumption 8, there exist $0 \leq \mathcal{L}, \sigma^2 < \infty$ such that almost surely

$$\forall k \in \mathbb{N}, \quad \mathbb{E}_k \left[ \|g(\theta_k, \xi_{k+1})\|_2^2 \right] \leq 2\mathcal{L} (F(\theta_k) - F^\star) + \sigma^2.$$

Inject these bounds in the previous inequality and substract $F(\theta^\star)$ on both sides to have

$$\mathbb{E}_k \left[ F(\theta_{k+1}) - F^\star \right] \leq (1 + L\mathcal{L}u_{k+1}^2)(F(\theta_k) - F^\star) - v_{k+1} \|\nabla F(\theta_k)\|_2^2 + (L/2)u_{k+1}^2 \sigma^2.$$

Introduce $V_k = F(\theta_k) - F^\star, W_k = v_{k+1} \|\nabla F(\theta_k)\|_2^2, a_k = L\mathcal{L}u_{k+1}^2$ and $b_k = (L/2)u_{k+1}^2 \sigma^2$. These four random sequences are non-negative $\mathcal{F}_k$-measurable sequences with $\sum_k a_k < \infty$ and $\sum_k b_k < \infty$ almost surely. Moreover we have

$$\forall k \in \mathbb{N}, \quad \mathbb{E}\left[ V_{k+1} | \mathcal{F}_k \right] \leq (1 + a_k)V_k - W_k + b_k.$$

We can apply Robbins-Siegmund Theorem 6 to have

$$(a) \sum_{k \geq 0} W_k < \infty \ a.s. \qquad (b) \ V_k \xrightarrow{a.s.} V_\infty, \mathbb{E}\left[ V_\infty \right] < \infty. \qquad (c) \ \sup_{k \geq 0} \mathbb{E}\left[ V_k \right] < \infty.$$

Therefore we have the almost sure convergence of the series $\sum v_{k+1} \|\nabla F(\theta_k)\|_2^2$ which, given that $\limsup_k \nu_k/\mu_k$ exists, implies that $\sum u_{k+1} \|\nabla F(\theta_k)\|_2^2$ is finite. Hence we obtain (i) in Lemma 1. We now show that (ii) in Lemma 1 is also valid. The term of interest is a sum of martingale increments. The quadratic variation is given by

$$\sum_{t \geq 1} \gamma_t^2 \mathbb{E}_t[\|C_{t-1}(g_{t-1}(\omega) - \nabla f(\theta_{t-1}(\omega)))\|_2^2] \leq \sum_{t \geq 1} \gamma_t^2 \nu_t^2 \mathbb{E}_t[\|(g_{t-1}(\omega) - \nabla F(\theta_{t-1}(\omega)))\|_2^2]$$

$$\leq \sum_{t \geq 1} \gamma_t^2 \nu_t^2 \mathbb{E}_t[\|g_{t-1}(\omega)\|_2^2]$$

$$\leq \sum_{t \geq 1} \gamma_t^2 \nu_t^2 (2\mathcal{L}(F(\theta_{t-1}) - F^\star) + \sigma^2).$$

Now we can use that $V_k = F(\theta_k) - F^\star \xrightarrow{a.s.} V_\infty$ (which was deduced from Robbins-Siegmund Theorem) to obtain that the previous series converges. Invoking Theorem 2.17 in Hall & Heyde (1980), we obtain (ii) in Lemma 1. Furthermore we can prove that $\theta_{k+1} - \theta_k \to 0$ almost surely and in $L^2$. Indeed, we have

$$\mathbb{E}\left[\|\theta_{k+1} - \theta_k\|_2^2\right] = \mathbb{E}\left[\|\gamma_{k+1} C_k g(\theta_k, \xi_{k+1}\|_2^2\right] \le u_{k+1}^2 \left(2\mathcal{L}\left(F(\theta_k) - F^\star\right) + \sigma^2\right).$$

In virtue of the almost sure convergence of $V_k = F(\theta_k) - F^\star$, the last term in parenthesis is upper bounded by a constant so that in view of the convergence of $\sum u_{k+1}^2$, we have the convergence of the series $\sum \mathbb{E}\left[\|\theta_{k+1} - \theta_k\|_2^2\right]$. We then deduce that $\mathbb{E}\left[\|\theta_{k+1} - \theta_k\|_2^2\right] \to 0$ and $\sum \left[\|\theta_{k+1} - \theta_k\|_2^2\right] < +\infty$ almost surely. In particular, $\theta_{k+1} - \theta_k \to 0$ in $L^2$ and almost surely. The last point follows from the fact that, for every $\delta > 0$,

$$\lim_{n \to \infty} \mathbb{P}\left(\sup_{k \ge n} \|\theta_{k+1} - \theta_k\| \ge \delta\right) \le \delta^{-2} \lim_{n \to \infty} \sum_{k \ge n} \mathbb{E}\left[\|\theta_{k+1} - \theta_k\|_2^2\right] = 0.$$

### A.3 Proof of Corollary 2

First observe that since $F$ is coercive, the convergence of $(F(\theta_k))$ obtained by Robbins-Siegmund theorem implies that the sequence of iterates $(\theta_k)_{k \ge 0}$ remains in a compact subset $\mathcal{K} \subset \mathbb{R}^d$. Let $\varepsilon > 0$. Since $\theta \mapsto d(\theta, \mathcal{S})$ is continuous, the set $\mathcal{D}(\varepsilon) = \{\theta \in \mathbb{R}^d : d(\theta, \mathcal{S}) \ge \varepsilon\}$ is closed and the set $\mathcal{K}(\varepsilon) = \mathcal{K} \cap \mathcal{D}(\varepsilon)$ is compact. On this set, the map $\theta \mapsto \|\nabla F(\theta)\|_2$ is stricly positive and there exists $\eta_\varepsilon > 0$ such that: $\theta \in \mathcal{K}(\varepsilon) \Rightarrow \|\nabla F(\theta)\|_2 > \eta_\varepsilon$. Thus, $\mathbb{P}(\theta \in \mathcal{K}(\varepsilon)) \le \mathbb{P}(\|\nabla F(\theta)\|_2 > \eta_\varepsilon)$ and this last quantity goes to zero which proves the convergence in probability $d(\theta_k, \mathcal{S}) \to 0$. Actually the almost sure convergence $\nabla F(\theta_k) \to 0$ implies the convergence of the distances. Define $A_k(\varepsilon) = \{\omega : \theta_k(\omega) \in \mathcal{K}(\varepsilon)\}$ and $B_k(\varepsilon) = \{\omega : \|\nabla F(\theta_k(\omega))\|_2 > \eta_\varepsilon\}$. We have $A_k(\varepsilon) \subset B_k(\varepsilon)$ then $\cup_{n \ge 1} \cap_{k \ge n} A_k(\varepsilon) \subset \cup_{n \ge 1} \cap_{k \ge n} B_k(\varepsilon)$. Conclude by using the almost sure convergence $\mathbb{P}(\cup_{n \ge 1} \cap_{k \ge n} B_k(\varepsilon)) = 0$ for each $\varepsilon > 0$. If $\mathcal{S}$ is finite, it is in particular a compact set so the distance is attained for every $k \ge 0$, $d(\theta_k, \mathcal{S}) = \min_{s \in \mathcal{S}} d(\theta_k, s) \to 0$. Since $\theta_{k+1} - \theta_k \to 0$, the sequence of iterates can only converge to a single point of $\mathcal{S}$.

## B Practical procedure

For the sake of completeness, the aim of this Section is to derive a feasible procedure that achieves the optimal asymptotic variance described in Corollary 1. First, we present a practical way to compute the *conditioning* matrix $C_k$ and then we show that the resulting algorithm satisfies the high-level conditions of Theorem 2. This method is considered in a numerical illustration along with a novel variant of AdaGrad.

### B.1 Construction of the conditioning matrix $C_k$

Similarly to the unavailability of gradients, one may not have access to values of the Hessian matrix but only stochastic versions of it (see details in numerical experiments below). As a consequence, we consider the following framework which involves random Hessian matrices. As for gradients, a policy $(P_k')_{k \ge 0}$ is used at each iteration to produce random Hessians through $H(\theta_k, \xi_{k+1}')$ with $\xi_{k+1}' \sim P_k'$.

**Assumption 10** (Unbiased and bounded Hessians). *The Hessian generator $H : \mathbb{R}^d \times S \to \mathbb{R}^{d \times d}$ is uniformly bounded around the minimizer and is such that for all $\theta \in \mathbb{R}^d$, $H(\theta, \cdot)$ is measurable and we have:* $\forall k \ge 0, \quad \mathbb{E}\left[H(\theta_k, \xi_{k+1}')|\mathcal{F}_k\right] = \nabla^2 F(\theta_k)$.

An estimate of the Hessian matrix $H = \nabla^2 F(\theta^\star)$ is now introduced as the weighted average

$$\Phi_k = \sum_{j=0}^{k} \omega_{j,k} H(\theta_j, \xi_{j+1}') \quad \text{with} \quad \sum_{j=0}^{k} \omega_{j,k} = 1. \tag{22}$$

The previous estimate has two advantages. First, thanks to averaging, the noise associated to each evaluation $H(\theta_j, \xi_{j+1}')$ will eventually vanished due to the sum of martingale increments. Second, the weights $\omega_{j,k}$ may help to give more importance to most recent iterates. In the idea that $\theta_k$ lies near $\theta^\star$ eventually, it might be helpful to reduce the bias when estimating $H = \nabla^2 F(\theta^\star)$.

**Proposition 2.** *Let* $(\Phi_k)_{k \geq 0}$ *be obtained by* (22). *Suppose that Assumptions 3 and 10 are fulfilled and that* $\theta_k \to \theta^\star$ *a.s. If* $\sup_{0 \leq j \leq k} \omega_{j,k} = O(1/k)$, *then we have* $\Phi_k \to H = \nabla^2 F(\theta^\star)$ *a.s.*

A natural choice is to take equal weights $\omega_{j,k} = (k+1)^{-1}$. However, since the last iterates are more likely to bring more relevant information through their Hessian estimates, we advocate the use of adaptive weights of the form $\omega_{j,k} \propto \exp(-\eta\|\theta_j - \theta_k\|_1)$ with a parameter $\eta \geq 0$ that recovers equal weights with $\eta = 0$. These two weights sequences satisfy the assumption of Proposition 2. They are considered in the numerical illustration of the next Section. While inverting $\Phi_k$ would produce a simple estimate of $H^{-1}$, such an approach might result in a certain instability in practice caused by large jumps towards wrong directions (large eigenvalues) or a too restrictive visit along other components (vanishing eigenvalues). To overcome this issue, we rely on the following filter which clamps the eigenvalues of a symmetric matrix. For any symmetric matrix $S$ and two positive numbers $0 < a < b$, denote by $S[a, b]$ the associated matrix where all the eigenvalues are clamped to $[a, b]$, *i.e.*, any eigenvalue $\lambda$ of $S$ is modified as $\lambda \leftarrow \max\{a, \min\{\lambda, b\}\}$.

Let $(\lambda_k^{(m)})_{k \geq 1}$ and $(\lambda_k^{(M)})_{k \geq 1}$ be two sequence of positive numbers such that $\lambda_k^{(m)} \leq \lambda_k^{(M)}$ for all $k \geq 1$. Define the matrices

$$\forall k \in \mathbb{N}, \quad C_k = \left( \Phi_k[(\lambda_{k+1}^{(M)})^{-1}, (\lambda_{k+1}^{(m)})^{-1}] \right)^{-1}. \tag{23}$$

Such a definition guarantees two properties. First, $C_k \in \mathcal{S}_d^{++}(\mathbb{R})$ with $\lambda_{k+1}^{(m)} I_d \preceq C_k \preceq \lambda_{k+1}^{(M)} I_d$. Second, in virtue of Proposition 2, $\Phi_k \to H$ a.s. so that, as soon as $(\lambda_k^{(m)})_{k \geq 1}$ and $(\lambda_k^{(M)})_{k \geq 1}$ go to 0 and $+\infty$ respectively, the matrix $C_k$ converges almost surely to $H^{-1}$ (as recommended by Corollary 1). Therefore, we obtain a feasible procedure leading to asymptotic optimality.

**Theorem 5** (Asymptotic optimality of the iterates). *Let* $(\theta_k)_{k \geq 0}$ *be obtained by conditioned SGD* (2) *with* $\gamma_k = 1/k$, $\Phi_k$ *defined by* (22), $\lambda_k^{(m)} \to 0, \lambda_k^{(M)} \to +\infty$ *and* $C_k$ *given by* (23). *Suppose that Assumptions 1 to 9 are fulfilled and* $\sup_{0 \leq j \leq k} \omega_{j,k} = O(1/k)$. *We have*

$$\sqrt{k}(\theta_k - \theta^\star) \rightsquigarrow \mathcal{N}(0, H^{-1}\Gamma H^{-1}), \qquad as \ k \to \infty.$$

This algorithm is theoretically asymptotically optimal. However in practice, adaptive gradient methods described in Table 1 have become the workhorse for training deep learning models as they take advantage of low rank-approximations and diagonal scalings. Interestingly, the *conditioned* matrices involved in these methods are linked to gradient estimates and thus to covariance matrices $\Gamma_k$ (see Assumption 4) rather than the Hessian H. Indeed, since $\theta^\star \in \mathcal{S}$, we have for the limiting covariance $\Gamma = \mathbb{E}_\xi[g(\theta^\star, \xi)g(\theta^\star, \xi)^\top]$. Consider a variant of AdaGrad which accumulates the average gradients $G_k = \delta I + (1/k)\sum_{i=1}^k g_i g_i^\top$ and $C_k = G_k^{-1/2}$. Averaging allows to anneal the stochastic noise of the gradient estimate. By the law of large numbers, the limiting matrix in our Theorem 2 will be $C = (\Gamma + \delta I)^{-1/2}$.

## B.2 Numerical illustration

Consider the empirical risk minimization framework applied to Generalized Linear Models. Given a data matrix $X = (x_{i,j}) \in \mathbb{R}^{n \times d}$ with labels $y \in \mathbb{R}^n$ and a regularization parameter $\lambda > 0$, we are interested in solving $\min_{\theta \in \mathbb{R}^d} F(\theta)$ with

$$F(\theta) = \frac{1}{n}\sum_{i=1}^n f_i(\theta), \quad f_i(\theta) = \mathcal{L}(x_i^\top\theta, y_i) + \lambda\Omega(\theta),$$

$\mathcal{L} : \mathbb{R} \times \mathbb{R} \to \mathbb{R}$ is smooth loss function and $\Omega : \mathbb{R}^d \to \mathbb{R}_+$ is a smooth convex regularizer chosen as Tikhonov regularization $\Omega(\theta) = \frac{1}{2}\|\theta\|_2^2$. The gradient and Hessian of each component $f_i$ are given for all $i = 1, \ldots, n$ by

$$\nabla f_i(\theta) = \mathcal{L}'(x_i^\top\theta, y_i)x_i + \lambda\theta$$
$$\nabla^2 f_i(\theta) = \mathcal{L}''(x_i^\top\theta, y_i)x_i x_i^\top + \lambda I_d,$$

where $\mathcal{L}'(\cdot,\cdot)$ and $\mathcal{L}''(\cdot,\cdot)$ are the first and second derivative of $\mathcal{L}(\cdot,\cdot)$ with respect to the first argument. Consider two well-known losses, namely least-squares and logistic. These losses are respectively associated to the Ridge regression problem with $y \in \mathbb{R}^n$ and the binary classication task with $y \in \{-1,+1\}^n$. The regularization parameter is set to the classical value $\lambda = 1/n$. Denote by $\sigma(z) = 1/(1+\exp(-z))$ the sigmoid function, we have the following closed-form equations



(Ridge Regression)

$$\begin{cases} \mathcal{L}(x_i^\top \theta, y_i) & = \frac{1}{2}(y_i - x_i^\top \theta)^2 \\ \mathcal{L}'(x_i^\top \theta, y_i) & = x_i^\top \theta - y_i \\ \mathcal{L}''(x_i^\top \theta, y_i) & = 1 \end{cases}$$

(Logistic Regression)

$$\begin{cases} \mathcal{L}(x_i^\top \theta, y_i) & = \log(1 + \exp(-y_i x_i^\top \theta)) \\ \mathcal{L}'(x_i^\top \theta, y_i) & = \sigma(x_i^\top \theta) - y_i \\ \mathcal{L}''(x_i^\top \theta, y_i) & = \sigma(x_i^\top \theta)(1 - \sigma(x_i^\top \theta)) \end{cases}$$



As stated in Example 1 of Section 2, stochastic versions of both the gradient and the Hessian of the objective $F$ can be easily computed using only a batch $B \subset \{1, \ldots, n\}$ of data and $\nabla_B F(\theta) = \sum_{i \in B} \nabla f_i(\theta)/|B|$ (resp. $\nabla_B^2 F(\theta) = \sum_{i \in B} \nabla^2 f_i(\theta)/|B|$) for the gradient (resp. Hessian) estimate. Note that these random generators meet Assumptions 1 and 10 as they produce unbiased estimates of the gradient and the Hessian matrix respectively.

For the sake of completeness and illustrative purposes, we compare the performance of classical stochastic gradient descent (sgd) and the *conditionned* variant (csgd) presented in Appendix B where the matrix $\Phi_k$ is an averaging of past Hessian estimates as given in Equation (22). We shall compare equal weights $\omega_{j,k} = (k+1)^{-1}$ and adaptive weights $\omega_{j,k} \propto \exp(-\eta \|\theta_j - \theta_k\|_1)$ with $\eta > 0$ to give more importance to Hessian estimates associated to iterates which are closed to the current point. Furthermore, for computational reason, we consider a novel adaptive stochastic first-order method which is a variant of Adagrad.

Starting from the null vector $\theta_0 = (0, \ldots, 0) \in \mathbb{R}^d$, we use optimal learning rate of the form $\gamma_k = \alpha/(k+k_0)$ (Bottou et al., 2018) and set $\lambda_k^{(m)} \equiv 0, \lambda_k^{(M)} = \Lambda\sqrt{k}$ in the experiments where $\gamma, k_0$ and $\Lambda$ are tuned using a grid search. The means of the optimality ratio $k \mapsto [F(\theta_k) - F(\theta^\star)]/[F(\theta_0) - F(\theta^\star)]$, obtained over 100 independent runs, are presented in Figures below.

**Methods in competition.** The different methods in the experiments are:

- *sgd*: standard stochastic gradient descent.

- *sgd_avg*: Polyak-averaging stochastic gradient descent , with a burn-in period ($n_0 = 15$ for $d = 20$ and $n_0 = 30$ for $d = 100$) to avoid the poor performance of bad initialization.

- *csgd($\eta = 0$)* and *csgd($\eta > 0$)*: *conditioned* stochastic gradient descent methods with equal and adaptive weights where the matrix $\Phi_k$ is an averaging of past Hessian estimates as given in Equation (22).

- *adafull_avg*: The variant of Adagrad presented in Appendix B where the gradient matrix $G_k$ is updated as an average $G_k = \delta I + (1/k)\sum_{i=1}^k g_i g_i^\top$ and $C_k = G_k^{-1/2}$ instead of the cumulative sum provided in the literature of Adagrad. Note that averaging here allows to anneal the stochastic noise whereas classical versions of Adagrad often rely on true gradients and may use cumulative sums. The parameter $\delta$ is also tuned using a grid search.

We focus on Ridge regression on simulated data with $n = 10,000$ samples in dimensions $d \in \{20; 100\}$. Stochastic gradient methods are known to greatly benefit from mini-batch instead of picking a single random sample when computing the gradient estimate. We use a batch-size equal to $|B| = 16$. In Figure 1, we can see that *conditioned* SGD outperforms standard SGD. Furthermore, adaptive weights ($\eta > 0$) improve the convergence speed of *conditioned* SGD methods. Interestingly, the novel approach *adafull_avg* offers great performance at a cheap computing cost. Indeed, the update of $C_{k+1}$ relies on the inverse of an average. This operation can be carried out in an efficient way thanks to Woodbury matrix identity.

**Real-world data.** We now turn our attention to real-world data and consider again the Ridge regression problem on the following datasets: *Boston Housing dataset* (Harrison Jr & Rubinfeld, 1978) ($n = 506; d = 14$) and *Diabetes dataset* (Dua & Graff, 2017) ($n = 442; d = 10$).

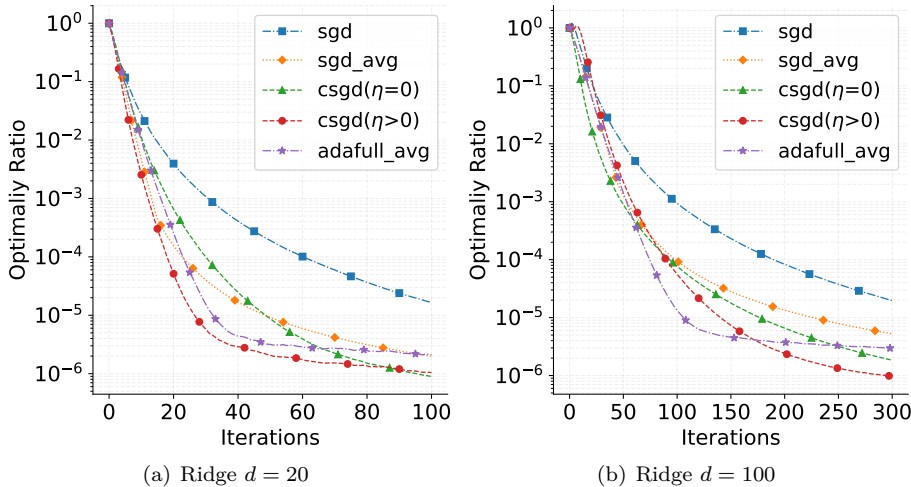

Figure 1: Optimality ratio $k \mapsto [F(\theta_k) - F(\theta^\star)]/[F(\theta_0) - F(\theta^\star)]$ for Ridge regression in dimension $d \in \{20; 100\}$.

- *Boston Housing dataset* (Harrison Jr & Rubinfeld, 1978): This dataset contains information collected by the U.S Census Service concerning housing in the area of Boston Mass. It contains $n = 506$ samples in dimension $d = 14$.

- *Diabetes dataset* (Dua & Graff, 2017): Ten baseline variables, age, sex, body mass index, average blood pressure, and six blood serum measurements were obtained for each of $n = 442$ diabetes patients, as well as the response of interest, a quantitative measure of disease progression one year after baseline.

The means of the optimality ratio $k \mapsto [F(\theta_k) - F(\theta^\star)]/[F(\theta_0) - F(\theta^\star)]$, obtained over 100 independent runs, are presented in Figure 2. Once again, the *conditioned* SGD methods offer better performance than plain SGD. For these datasets, it is the *conditioning* matrix with adaptive weights as given in Equation (22) which presents the best results.

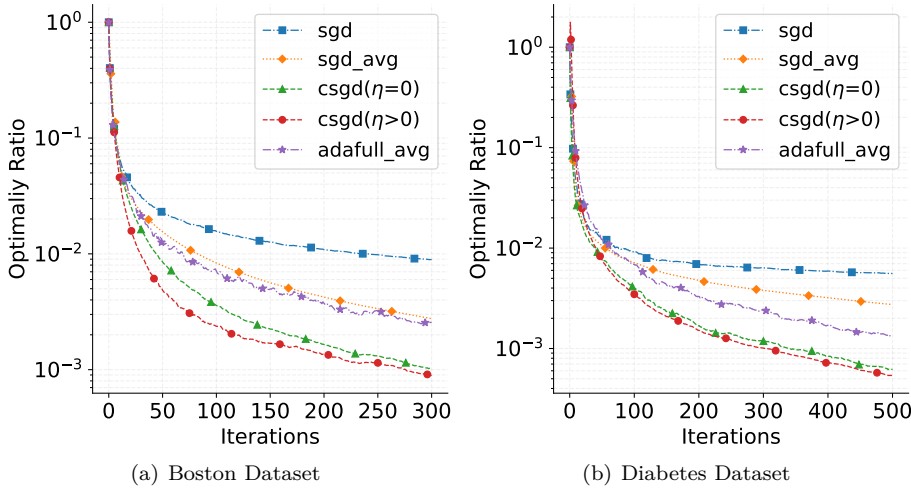

Figure 2: Optimality ratio $k \mapsto [F(\theta_k) - F(\theta^\star)]/[F(\theta_0) - F(\theta^\star)]$ for Ridge regression on datasets Boston and Diabetes.

## C  Auxiliary results

### C.1  Robbins-Siegmund Theorem

**Theorem 6.** *(Robbins & Siegmund (1971)) Consider a filtration $(\mathcal{F}_n)_{n\geq 0}$ and four sequences of random variables$(V_n)_{n\geq 0}, (W_n)_{n\geq 0}, (a_n)_{n\geq 0}$ and $(b_n)_{n\geq 0}$ that are adapted and non-negative. Assume that almost surely $\sum_k a_k < \infty$ and $\sum_k b_k < \infty$. Assume moreover that $\mathbb{E}[V_0] < \infty$ and for all $n \in \mathbb{N}$, $\mathbb{E}[V_{n+1}|\mathcal{F}_n] \leq (1 + a_n)V_n - W_n + b_n$. Then it holds*

$$(a) \sum_k W_k < \infty \ a.s. \qquad (b) \ V_n \xrightarrow{a.s.} V_\infty, \mathbb{E}[V_\infty] < \infty. \qquad (c) \ \sup_{n\geq 0} \mathbb{E}[V_n] < \infty.$$

### C.2  Auxiliary lemmas

**Lemma 2.** *Let $(u_n)_{n\geq 1}, (v_n)_{n\geq 1}$ and $(\gamma_n)_{n\geq 1}$ be non-negative sequences such that $\gamma_n \to 0$ and $\sum_n \gamma_n = +\infty$. Assume that there exists a real number $m \geq 1$ and $j \geq 1$ such that for all $n \geq j$, $u_n \leq (1-\gamma_n)^m u_{n-1} + \gamma_n v_n$. Then it holds that $\limsup_{n\to+\infty} u_n \leq \limsup_{n\to+\infty} v_n$.*

*Proof.* Denote $x_+ = \max(x, 0)$. One has $(x+y)_+ \leq x_+ + y_+$. Set $\varepsilon > 0$ and $v = \limsup_n v_n + \varepsilon$. Then there exists an integer $N \geq 1$ such that $(1 - \gamma_n)^m \leq (1 - \gamma_n)$ and $v_n < v$, i.e., $(v_n - v)_+ = 0$ for $n \geq N$. We have for large enough $n \geq N \vee j$,

$$u_n - v \leq (1 - \gamma_n)(u_{n-1} - v) + \gamma_n(v_n - v),$$

and taking the positive part gives

$$(u_n - v)_+ \leq (1 - \gamma_n)(u_{n-1} - v)_+ + \gamma_n(v_n - v)_+ = (1 - \gamma_n)(u_{n-1} - v)_+.$$

Since $\sum_n \gamma_n = +\infty$, this inequality implies that $(u_n - v)_+$ tends to zero, but this is true for all $\varepsilon > 0$ so $v$ is arbitrarily close to $\limsup_n v_n$ and the result follows. $\square$

**Lemma 3.** *Let $(\gamma_n)_{n\geq 1}$ be a non-negative sequence converging to zero, and $\lambda, m$ and $p$ three real numbers with $\lambda > 0, m \geq 1, p \geq 0$. Consider two non-negative sequences $(x_n), (\varepsilon_n)$ and an integer $j \geq 1$ such that*

$$\forall n \geq j, \quad x_n = (1 - \lambda\gamma_n)^m x_{n-1} + \gamma_n^{p+1}\varepsilon_n,$$

$$i.e., \quad x_n = \prod_{i=j}^n (1 - \lambda\gamma_i)^m x_{j-1} + \sum_{k=j}^n \gamma_k^{p+1}\left(\prod_{i=k+1}^n (1 - \lambda\gamma_i)^m\right)\varepsilon_k.$$

*The following holds*
* *if $\gamma_n = n^{-\beta}, \beta \in (1/2, 1)$, then for any $p$*

$$\limsup_{n\to+\infty} \frac{x_n}{\gamma_n^p} \leq \frac{1}{m\lambda}\limsup_{n\to+\infty}\varepsilon_n.$$

* *if $\gamma_n = 1/n$, then for any $p < m\lambda$*

$$\limsup_{n\to+\infty} \frac{x_n}{\gamma_n^p} \leq \frac{1}{m\lambda - p}\limsup_{n\to+\infty}\varepsilon_n.$$

*In particular, when $\varepsilon_n \to 0$ with $j = 1$ and $x_0 = 0$,*

$$\lim_{n\to+\infty} \sum_{k=1}^n \gamma_k \prod_{i=k+1}^n (1 - \lambda\gamma_i)^m \varepsilon_k = 0,$$

$$(m\lambda > 1) \lim_{n\to+\infty} \frac{1}{\gamma_n}\sum_{k=1}^n \gamma_k^2 \prod_{i=k+1}^n (1 - \lambda\gamma_i)^m \varepsilon_k = 0.$$

Before proving this result, note that if we consider $\gamma_n = \gamma/n^\beta$ then we can write

$$x_n = (1 - \lambda\gamma_n)^m x_{n-1} + \gamma_n^{p+1}\varepsilon_n = (1 - (\lambda\gamma)n^{-\beta})^m x_{n-1} + (n^{-\beta})^{p+1}(\gamma^{p+1}\varepsilon_n)$$

and apply the result with $\tilde{\lambda} = \gamma\lambda$ and $\tilde{\varepsilon}_n = \gamma^{p+1}\varepsilon_n$.

*Proof.* We apply Lemma 2 to the sequence $u_n = \frac{x_n}{\gamma_n^p}$. We have for all $n \geq j$,

$$u_n = \frac{1}{\gamma_n^p}\left((1 - \lambda\gamma_n)^m x_{n-1} + \gamma_n^{p+1}\varepsilon_n\right)$$

$$= \left(\frac{\gamma_{n-1}}{\gamma_n}\right)^p (1 - \lambda\gamma_n)^m u_{n-1} + \gamma_n\varepsilon_n$$

$$= \exp\left(p\log\left(\frac{\gamma_{n-1}}{\gamma_n}\right) + m\log(1 - \lambda\gamma_n)\right)u_{n-1} + \gamma_n\varepsilon_n.$$

Define

$$\lambda_n = \frac{1}{\gamma_n}\left(1 - \exp\left(p\log\left(\frac{\gamma_{n-1}}{\gamma_n}\right) + m\log(1 - \lambda\gamma_n)\right)\right),$$

so we get the recursion equation

$$\forall n \geq j, \quad u_n = (1 - \lambda_n\gamma_n)u_{n-1} + \lambda_n\gamma_n\frac{\varepsilon_n}{\lambda_n}.$$

• if $\gamma_n = n^{-\beta}, \beta \in (1/2, 1)$ then $1/\gamma_n - 1/\gamma_{n-1} \to 0$ and the ratio $\gamma_{n-1}/\gamma_n$ tends to 1 with

$$\log\left(\frac{\gamma_{n-1}}{\gamma_n}\right) = \left(\frac{\gamma_{n-1}}{\gamma_n} - 1\right)(1 + o(1)) = \gamma_{n-1}\left(\frac{1}{\gamma_n} - \frac{1}{\gamma_{n-1}}\right)(1 + o(1)) = o(\gamma_n).$$

Besides, $m\log(1 - \lambda\gamma_n) = -m\lambda\gamma_n + o(\gamma_n)$ when $n \to +\infty$ and we get

$$\lambda_n = \frac{1}{\gamma_n}\left[1 - \exp\left(-m\lambda\gamma_n + o(\gamma_n)\right)\right],$$

which implies that $\lambda_n$ converges to $m\lambda$. We conclude with Lemma 2.
• if $\gamma_n = 1/n$ then the ratio $\gamma_{n-1}/\gamma_n$ tends to 1 with

$$\log\left(\frac{\gamma_{n-1}}{\gamma_n}\right) = \log\left(1 + \frac{1}{n-1}\right) = \gamma_n + o(\gamma_n).$$

We still have $m\log(1 - \lambda\gamma_n) = -m\lambda\gamma_n + o(\gamma_n)$ when $n \to +\infty$ and therefore

$$\lambda_n = \frac{1}{\gamma_n}\left[1 - \exp\left((p - m\lambda)\gamma_n + o(\gamma_n)\right)\right],$$

which implies $\lambda_n$ converges to $(m\lambda - p)$ and we conclude in the same way. $\qquad\square$

**Lemma 4.** *Let* $A, B \in \mathcal{S}_d^{++}(\mathbb{R})$ *then the eigenvalues of* $AB$ *are real and positive with* $Sp(AB) \subset [\lambda_{\min}(A)\lambda_{\min}(B); \lambda_{\max}(A)\lambda_{\max}(B)]$.

*Proof.* Denote by $\sqrt{B}$ the unique positive square root of $B$. The matrix $AB$ is similar to the real symmetric positive definite matrix $\sqrt{B}A\sqrt{B}$. Therefore its eigenvalues are real and positive. Since $A \mapsto \lambda_{\max}(A)$ is a sub-multiplicative matrix norm on $\mathcal{S}_d^{++}(\mathbb{R})$, $\lambda_{\max}(AB) \leq \lambda_{\max}(A)\lambda_{\max}(B)$ which gives $\lambda_{\max}((AB)^{-1}) \leq \lambda_{\max}(A^{-1})\lambda_{\max}(B^{-1})$, *i.e.,* $\lambda_{\min}(AB)^{-1} \leq \lambda_{\min}(A)^{-1}\lambda_{\min}(B)^{-1}$, and finally $\lambda_{\min}(A)\lambda_{\min}(B) \leq \lambda_{\min}(AB)$. $\qquad\square$

**Lemma 5.** *Let $S \in \mathcal{S}_d^{++}(\mathbb{R})$ be a real symmetric positive definite matrix. Let $(\gamma_k)_{k \geq 1}$ be a positive decreasing sequence converging to 0 such that $\sum_k \gamma_k = +\infty$. Denote by $\lambda_m$ the smallest eigenvalue of $S$. It holds that there exists $j \geq 1$ such that for any $k > j$, all the eigenvalues of the real symmetric matrix $A_k = I - \gamma_k S$ are positive and we have*

$$\rho(\Pi_n) = \rho(A_n \dots A_1) \overset{n \to +\infty}{\longrightarrow} 0,$$

$$\forall k > j, \quad \rho(\Pi_{n,k}) = \rho(A_n \dots A_{k+1}) \leq \prod_{i=k+1}^{n} (1 - \gamma_i \lambda_m).$$

*Proof.* For any $k \in \mathbb{N}$, the eigenvalues of the real symmetric matrix $A_k = I - \gamma_k S$ are given by $Sp(A_k) = \{(1 - \gamma_k \lambda), \lambda \in Sp(S)\}$. Since $\gamma_k \to 0$, there exists $j \geq 1$ such that $\gamma_k \lambda_m < 1$ for all $k > j$. Therefore for any $k > j$, we have $Sp(A_k) \subset \mathbb{R}_+^\star$ and the largest eigenvalue is $\rho(A_k) = 1 - \gamma_k \lambda_m$. Since $\rho$ is a sub-multiplicative norm for real symmetric matrices, we get $\rho(\Pi_n) \leq \prod_{k=1}^{n} \rho(A_k) = \prod_{k=1}^{j} \rho(A_k) \prod_{k=j+1}^{n} \rho(A_k)$. The second product can be upper bounded with the convexity of exponential,

$$\prod_{k=j+1}^{n} \rho(A_k) = \prod_{k=j+1}^{n} (1 - \gamma_k \lambda_m) \leq \prod_{k=j+1}^{n} \exp(-\gamma_k \lambda_m) = \exp(-\lambda_m(\tau_n - \tau_j)) \overset{n \to +\infty}{\longrightarrow} 0.$$

Similarly we have for all $k > j, \rho(\Pi_{n,k}) \leq \prod_{i=k+1}^{n} \rho(A_i) \leq \prod_{i=k+1}^{n} (1 - \gamma_i \lambda_m)$. $\qquad\square$

**Lemma 6.** *Let $\gamma_n = \alpha n^{-\beta}$ with $\beta \in (1/2, 1]$ then it holds*

$$(\beta < 1) \sum_{k=1}^{n} \gamma_k \sim \frac{n \gamma_n}{1 - \beta} = \frac{\alpha}{1 - \beta} n^{1-\beta}, \quad (\beta = 1) \sum_{k=1}^{n} \gamma_k \sim \alpha \log(n).$$

*Proof.* By series-integral comprison, $\int_1^{n+1} t^{-\beta} dt \leq \sum_{k=1}^{n} k^{-\beta} \leq 1 + \int_1^n t^{-\beta} dt$. $\qquad\square$

**Theorem 7.** *(Delyon & Portier, 2021, Theorem 17)(Freedman inequality) Let $(X_j)_{1 \leq j \leq n}$ be random variables such that $\mathbb{E}[X_j | \mathcal{F}_{j-1}] = 0$ for all $1 \leq j \leq n$ then, for all $t \geq 0$ and $v, m > 0$,*

$$\mathbb{P}\left( \Big| \sum_{j=1}^{n} X_j \Big| \geq t, \max_{j=1,\dots,n} |X_j| \leq m, \sum_{j=1}^{n} \mathbb{E}\left[ X_j^2 \mid \mathcal{F}_{j-1} \right] \leq v \right) \leq 2 \exp\left( -\frac{t^2/2}{v + tm/3} \right).$$

**Lemma 7.** *Let $A \in \mathbb{R}^{n \times n}$ be a symmetric positive semi-definite matrix. Then for any $B \in \mathbb{R}^{m \times n}$, the matrix $BAB^\top \in \mathbb{R}^{m \times m}$ is symmetric positive semi-definite.*

*Proof.* First note that $(BAB^\top)^\top = (B^\top)^\top A^\top B^\top = BAB^\top$ because $A$ is symmetric. Then for any vector $x \in \mathbb{R}$, we have $x^\top (BAB^\top)x = (B^\top x)^\top A(B^\top x) \geq 0$ since $A$ is positive semi-definite. $\qquad\square$

**Proposition 3.** *(Khalil, 2002, Theorem 4.6) Let $H$ be a positive definite matrix and $\Gamma$ a symmetric positive definite matrix of same dimension. Then there exists a symmetric positive definite matrix $\Sigma$, unique solution of the Lyapunov equation $H\Sigma + \Sigma H^\top = \Gamma$, which is given by $\Sigma = \int_0^{+\infty} e^{-tH} \Gamma e^{-tH^\top} dt$.*

The results remains true if the matrix $\Gamma$ is only symmetric positive semi-definite: in that case the matrix $\Sigma$ is also symmetric positive semi-definite and is the solution of the Lyapunov equation.

## C.3 Additional propositions

This section gathers the proofs of Proposition 1 about the optimal choice for the *conditioning* matrix and of Proposition 2 about the almost sure convergence of the *conditioning* matrices.

**Proposition 1.** *The choice $C^\star = H^{-1}$ is optimal in the sense that $\Sigma_{C^*} \preceq \Sigma_C, \forall C \in \mathcal{C}_H$. Moreover, $\Sigma_{C^\star} = H^{-1} \Gamma H^{-1}$.*

*Proof.* Define $\Delta_C = \Sigma_C - H^{-1}\Gamma H^{-1}$ and check that $\Delta_C$ satisfies

$$\left(CH - I_d/2\right)\Delta_C + \Delta_C\left(CH - I_d/2\right)^\top = (C - H^{-1})\Gamma(C - H^{-1}).$$

Because $\Gamma$ is symmetric positive semi-definite, we have using Lemma 7 that the term on the right side is symmetric positive semi-definite. Therefore, in view of Proposition 3, we get that $\Delta_C$ is symmetric positive semi-definite $\Delta_C \succeq 0$ which implies $\Sigma_C \succeq H^{-1}\Gamma H^{-1}$ for all $C \in \mathcal{C}_H$. The equality is reached for $C^\star = H^{-1}$ with $\Delta_C = 0, \Sigma_{C^\star} = H^{-1}\Gamma H^{-1}$. $\qquad\square$

**Proposition** 2. *Let $(\Phi_k)_{k \geq 0}$ be obtained by (22). Suppose that Assumptions 3 and 10 are fulfilled and that $\theta_k \to \theta^\star$ almost surely . If $\sup_{0 \leq j \leq k} \omega_{j,k} = O(1/k)$, then we have $\Phi_k \to H = \nabla^2 F(\theta^\star)$ almost surely.*

*Proof.* We use the decomposition

$$\Phi_k - H = \sum_{j=0}^{k} \omega_{j,k}\left(\nabla^2 F(\theta_j) - H\right) + \sum_{j=0}^{k} \omega_{j,k}\left(H(\theta_j, \xi'_{j+1}) - \nabla^2 F(\theta_j)\right).$$

The continuity of $\nabla^2 F$ at $\theta^\star$ and the fact that $\theta_j \to \theta^\star$ a.s. implie that $\left\|\nabla^2 F(\theta_j) - H\right\| \to 0$ a.s. Since $\sup_{0 \leq j \leq k} \omega_{j,k} = O(1/k)$, there exists $a > 0$ such that

$$\left\|\sum_{j=0}^{k} \omega_{j,k}\left(\nabla^2 F(\theta_j) - H\right)\right\| \leq \frac{a}{k+1}\sum_{j=0}^{k}\left\|\nabla^2 F(\theta_j) - H\right\|,$$

which goes to 0 in virtue of Cesaro's Lemma, therefore $\lim_{k\to\infty}\sum_{j=0}^{k} \omega_{j,k}\left(\nabla^2 F(\theta_j) - H\right) = 0$. The second term is a sum of martingale increments and shall be treated with Freedman inequality and Borel-Cantelli Lemma. Introduce the martingale increments

$$\forall 0 \leq j \leq k, \quad X_{j+1,k} = \omega_{j,k}\left(H(\theta_j, \xi'_{j+1}) - \nabla^2 F(\theta_j)\right).$$

For a fixed $k$, we have $X_{j+1,k} = \left(x_{j+1}^{(i,l)}\right)_{1 \leq i,l \leq d}$ where we remove the index $k$ for the sake of clarity. Because the Hessian generator is unbiased, we have for all coordinates

$$\mathbb{E}\left[x_{j+1}^{(i,l)}|\mathcal{F}_j\right] = 0 \quad \text{for all } 0 \leq j \leq k.$$

By definition of the Hessian generator and using that $(\nabla^2 F(\theta_j))$ is bounded, we get that $\left\|H(\theta_j, \xi'_{j+1}) - \nabla^2 F(\theta_j)\right\| = O(1)$ for all $j \geq 0$. For any $b > 0$, consider the following event

$$\Omega_b = \left\{\sup_{k \geq 0} \max_{j=0,\ldots,k}(k+1)\left|x_{j+1}^{(i,l)}\right| \leq b\right\},$$

and note that since $\omega_{j,k} = O(1/k)$ we have $\mathbb{P}(\Omega_b) \to 1$ as $b \to \infty$. On this event, the martingale increments and the variance term are bounded as

$$\max_{j=0,\ldots,k}\left|x_{j+1}^{(i,l)}\right| \leq b(k+1)^{-1}, \quad \sum_{j=0}^{k}\mathbb{E}\left[\left(x_{j+1}^{(i,l)}\right)^2 \mid \mathcal{F}_j\right] \leq b^2(k+1)^{-1}.$$

Using Freedman inequality (Theorem 7), we have for all coordinates $i, l = 1, \ldots, d$,

$$\mathbb{P}\left(\left|\sum_{j=0}^{k} x_{j+1}^{(i,l)}\right| > \varepsilon, \Omega_b\right) \leq 2\exp\left(-\frac{\varepsilon^2(k+1)}{2b(b+\varepsilon)}\right).$$

The last term is the general term of a convergent series. Apply Borel-Cantelli Lemma (Borel, 1909) to finally get almost surely on $\Omega_b$ that $\lim_{k\to\infty}\sum_{j=0}^{k} x_{j+1}^{(i,l)} = 0$. Since $b > 0$ is arbitrary and $\mathbb{P}(\Omega_b) \to 1$ when

$b \to \infty$, we have almost surely $\lim_{k\to\infty} \sum_{j=0}^{k} x_{j+1}^{(i,l)} = 0$. This is true for all the coordinates of the martingale increments and therefore

$$\lim_{k\to\infty} \sum_{j=0}^{k} \omega_{j,k} \left( H(\theta_j, \xi'_{j+1}) - \nabla^2 F(\theta_j) \right) = 0 \text{ a.s.}$$

$\square$

## C.4 Auxiliary results on expected smoothness

The following Lemma gives sufficient conditions to meet the weak growth condition on the stochastic noise as stated in Assumption 8.

**Lemma 8.** *Suppose that for all $k \geq 1, \theta \in \mathbb{R}^d, F(\theta) = \mathbb{E}\left[f(\theta, \xi_k)|\mathcal{F}_{k-1}\right]$ with $\xi_k \sim P_{k-1}$. Assume that for all $\xi_k \sim P_{k-1}$, the function $\theta \mapsto f(\theta, \xi_k)$ is L-smooth almost surely and there exists $m \in \mathbb{R}$ such that for all $\theta \in \mathbb{R}^d, f(\theta, \xi_k) \geq m$. Then a gradient estimate is given by $g(\theta, \xi) = \nabla f(\theta, \xi)$ and the growth condition of Assumption 8 is satisfied with $\sigma^2 = 2L(F^\star - m)$ and*

$$\forall \theta \in \mathbb{R}^d, \forall k \in \mathbb{N}, \quad \mathbb{E}\left[\|g(\theta, \xi_k)\|_2^2|\mathcal{F}_{k-1}\right] \leq 2L\left(F(\theta) - F^\star\right) + \sigma^2.$$

*Proof.* For all $\xi_k \sim P_{k-1}$, Lipschitz continuity of the gradient $\theta \mapsto \nabla f(\theta, \xi_k)$ implies (see Nesterov (2013))

$$f(y, \xi_k) \leq f(\theta, \xi_k) + \langle \nabla f(\theta, \xi_k), y - \theta \rangle + (L/2)\|y - \theta\|_2^2.$$

Plug $y = \theta - (1/L)\nabla f(\theta, \xi_k)$ and use the lower bound $f(y, \xi_k) \geq m$ to obtain

$$\frac{1}{2L}\|\nabla f(\theta, \xi_k)\|_2^2 \leq f(\theta, \xi_k) - f(y, \xi_k) \leq f(\theta, \xi_k) - m,$$

which gives,

$$\|g(\theta, \xi_k)\|_2^2 \leq 2L\left(f(\theta, \xi_k) - f(\theta^\star, \xi_k)\right) + 2L\left(f(\theta^\star, \xi_k) - m\right)$$

and conclude by taking the conditional expectation with respect to $\mathcal{F}_{k-1}$. $\square$

The next Lemma links our weak growth condition with the notion of expected smoothness as introduced in Gower et al. (2019). In particular, this notion can be extended to our general context where the sampling distribution can evolve through the stochastic algorithm.

**Lemma 9.** *(Expected smoothness) Assume that with probability one,*

$$\sup_{k \geq 1} \sup_{x \neq x^\star} \frac{\mathbb{E}\left[\|g(\theta, \xi_k) - g(\theta^\star, \xi_k)\|_2^2|\mathcal{F}_{k-1}\right]}{F(\theta) - F^\star} < \infty \quad and \quad \sup_{k \geq 1} \mathbb{E}\left[\|g(\theta^\star, \xi_k)\|_2^2|\mathcal{F}_{k-1}\right] < \infty.$$

*Then there exist $0 \leq \mathcal{L}, \sigma^2 < \infty$ such that*

$$\forall \theta \in \mathbb{R}^d, \forall k \in \mathbb{N}, \quad \mathbb{E}\left[\|g(\theta, \xi_k)\|_2^2|\mathcal{F}_{k-1}\right] \leq 2\mathcal{L}\left(F(\theta) - F^\star\right) + 2\sigma^2.$$

*Proof.* For all $\theta \in \mathbb{R}^d$ and all $k \in \mathbb{N}$, we have

$$\|g(\theta, \xi_k)\|_2^2 = \|g(\theta, \xi_k) - g(\theta^\star, \xi_k) + g(\theta^\star, \xi_k)\|_2^2$$
$$\leq 2\|g(\theta, \xi_k) - g(\theta^\star, \xi_k)\|_2^2 + 2\|g(\theta^\star, \xi_k)\|_2^2.$$

Using the expected smoothness, with probability one, there exists $0 \leq \mathcal{L} < \infty$ such that

$$\mathbb{E}\left[\|g(\theta, \xi_k) - g(\theta^\star, \xi_k)\|_2^2|\mathcal{F}_{k-1}\right] \leq \mathcal{L}\left(F(\theta) - F^\star\right).$$

Since the noise at optimal point is almost surely finite there exists $0 \leq \sigma^2 < \infty$ such that

$$\mathbb{E}\left[\|g(\theta^\star, \xi_k)\|_2^2|\mathcal{F}_{k-1}\right] \leq \sigma^2,$$

which allows to conclude by taking the conditional expectation. $\square$

