# OpenReview forum: "Asymptotic Analysis of Conditioned Stochastic Gradient Descent"
_TMLR — Accepted by TMLR_

### Review · Reviewer_bWM3 · 2023-06-18

**Summary Of Contributions:**

This paper considers the conditioned stochastic gradient descent and provides the weak convergence for this method for the non-convex objective functions.

**Audience:**

Yes

**Broader Impact Concerns:**

No broader impact concerns.

**Claims And Evidence:**

Yes

**Requested Changes:**

Please bring up the practical implication of this analysis to the main part of the paper and add more examples about where CSGD can be utilized for ML models along side showing that the assumptions of the paper hold for those models.

**Strengths And Weaknesses:**

Strength: All the prooves are clear to understand and easy to follow.

Weakness: The practical implication of the provided analysis for the machine learning community is not discussed enough. The numerical evaluations are just for the convex settings. Also, assumption 5 may not hold for all the stationary points of a non-convex objective function.

---

> ### Author Response · Authors · 2023-07-06
> **Response to Reviewer bWM3 (Part 1)**
>
> We thank the reviewer for his or her feedback. Please find some detailed responses to review points below, as well as a revised manuscript with tracked changes in red.
>
> **General implications**
>
> CSGD methods often aim to improve the efficiency and convergence properties of the traditional SGD algorithm. By conditioning the update rule on additional information, these methods can potentially mitigate issues such as poorly normalized objective (when the objective is multiplied by some factor, one should expect that the algorithm does not change) and high variance gradients, and also to favor some direction compared to others. The main practical implications of CSGD can include:
>
> - *Normalization*: CSGD techniques such as diagonal scalings (Adagrad, RMSprop) allow the algorithm to be independent of any normalization of the objective, which in practice can help for fine-tuning the algorithm leading to better performance. In some models this can be used to perform feature scaling or normalization by applying a matrix multiplication to the gradients (Adagrad). This helps ensure that the updates are proportional to the scales of the corresponding features, which can improve the convergence behavior and stability of the optimization process.
>
> - *Minimal variance*: stochastic second-order methods (Quasi-Newton, LBFGS) may help speed up the convergence of machine learning models, making them more efficient in terms of training time and computational resources.
>
> - *Handling high-dimensional problems*: In high-dimensional settings where the standard SGD algorithm may struggle, CSGD methods can offer improved convergence and stability, e.g. when using natural gradient methods to adapt to the geometry of the underlying data. This can be particularly relevant in tasks such as image recognition, natural language processing, and genomics.
>
> - *Data structure adaptation*: CSGD techniques can be used to perform adaptive selection of relevant coordinates for optimization and take advantage of particular data structures. If there is prior knowledge or information about the problem domain or the relationships between features, e.g. in overparameterized neural networks, CSGD can be used to incorporate this knowledge into the gradient updates. By premultiplying the gradients with a matrix that encodes the prior knowledge, the optimization process can be guided towards solutions that align with the known constraints or relationships. For instance, by using diagonal conditioning matrices, one can recover the framework of stochastic coordinate descent with adaptive sampling as in
>
> Leluc, R., & Portier, F. (2022). Sgd with coordinate sampling: Theory and practice. The Journal of Machine Learning Research, 23(1), 15470-15516.
>
> &#8594; In the revised version of our paper, we bring that up in the first paragraph of the introduction by mentioning clearly the aim of the different conditioning methods.
>
> **Insights of Weak convergence**
>
> By applying a weak convergence theorem, such as the Central Limit Theorem given in the paper, one can gain valuable insights into the optimization process and in understanding the uncertainty associated with the iterates' distribution. Some specific applications and implications of the main result of the paper include:
>
> - *Model Selection and Comparison*: The weak convergence theorem-based analysis can also be applied to compare different variants of Conditioned SGD or to select the most suitable matrix for conditioning the gradients. By comparing the convergence behaviors or distributional properties of different sequences of iterates, one can assess which variant or conditioning matrix yields better optimization performance.
>
> - *Confidence Interval Estimation and Hypothesis Testing*: Once the convergence behavior is established, weak convergence theorems enable the construction of confidence intervals for estimating the distribution parameters of the iterates. It also allows us to perform hypothesis tests on the distribution parameters of the iterates. By applying the CLT, one can estimate the mean, variance, or other moments of the iterates' distribution and construct confidence intervals around these estimates. These intervals quantify the uncertainty associated with the estimated parameters. For instance, the following paper
>
> Zhu, W., Chen, X., & Wu, W. B. (2023). Online covariance matrix estimation in stochastic gradient descent. Journal of the American Statistical Association, 118(541), 393-404.
>
> provides a method to estimate the covariance matrix of average SGD and then use this estimate to build confidence intervals.
>
> &#8594; In the revised version, we consider estimation of the covariance matrix within the particular framework of variational inference as detailed below.

---

> ### Author Response · Authors · 2023-07-06
> **Response to Reviewer bWM3 (Part 2)**
>
> **Novel section on CSGD applications**
>
> In the revised version of our paper we added a new section (Section 4) about variational inference using importance sampling following the recent paper:
>
> Jerfel, G., Wang, S., Wong-Fannjiang, C., Heller, K. A., Ma, Y., & Jordan, M. I. (2021, December). Variational refinement for importance sampling using the forward kullback-leibler divergence. In Uncertainty in Artificial Intelligence (pp. 1819-1829). PMLR.
>
> The purpose of this new section is to demonstrate the effectiveness of the CSGD approach in the specific context of sampling from a target distribution, given only the ability to evaluate its density. By employing the Forward KL (Kullback-Leibler) risk measure, as recommended in the above paper, in combination with our CSGD framework, we achieve the well-known Cramér-Rao bound. This bound represents the optimal variance achievable when estimating parameters from a model based on observed data from the target distribution. In addition, the approach proposed allows estimating the limiting covariance matrix iteratively.
>
> In this new section, we start by establishing the standard statistical setting, allowing us to define the Fisher information lower bound. Subsequently, we introduce the Variational inference problem, where we demonstrate that our CSGD approach yields the same variance as previously mentioned. This initial finding paves the way for comparing alternative approaches that rely on different divergence measures.

---

### Review · Reviewer_3EKA · 2023-06-19

**Summary Of Contributions:**

This paper provides an asymptotic analysis of SGD, in terms of central limit theorems, for the case where the SGD is preconditioned. The result is novel.

**Audience:**

Yes

**Broader Impact Concerns:**

theory work, no broader impact.

**Claims And Evidence:**

Yes

**Requested Changes:**

I am not sure about the paper's contributions' significance, I have the following questions/suggestions and open to discussion:

1- I think the authors can make it clearer in their case policy P_k has an explicit link to sampling patterns of data in Example 1. More generally, the "policy" seems to be sampling distribution that computes the stochastic gradients. It is a bit of a different terminology -- but would be good to clarify this.

2- For A2-A5: I would appreciate some detailed explanation for the intuition in the assumptions, optimally in the form of remarks.

For example, A2 is standard learning rate decay but known to fail to work in practice.

Why A3 is important and what does it "intuitively" bring?

Same for A5, a thorough explanation of the assumption would be useful. This is assuming that the MSE (more correctly 2+\delta power of the error in the gradient) of the stochastic gradients are bounded "around the stationary point". Is it a mild assumption or a strong assumption? How does it compare to usual bounded variance (or MSE) assumption in SGD?

Also a discussion about how these relates to standard assumptions for SGD (namely, \mu-strong convexity + L-smooth gradients) should be discussed. Are these weaker or stronger?

3- I found sketch of the proof a bit unhelpful -- not clarifying a lot the general idea. Also it would be good to cite here the exact section in Appendix where the proof can be found.

4- About Prop. 1 and Corollary 1: I think the results here could be made a bit more specific given the setting. While the optimal choice of the preconditioning matrix is the Hessian, this is often approximated in practice. What kind of approximation errors can be accounted for theory in this paper to hold? Is there a structure to this error? If the preconditioned matrix is the perturbed version (an approximation) of the Hessian, can theory say anything approximate here? I think this would be useful direction to enhance this paper and demonstrate the utility of the introduced result.

5- The last part of the paper seems to proceed under stronger assumptions (A6 onwards) to prove almost sure convergence. As I said above, it is also important to contrast these with the A2-A5 on what/how they differ and compare.

**Strengths And Weaknesses:**

General remarks: The paper is cleanly written and reads well -- with interesting content. I also found the part related to the review and summary of existing methods well written. I found the overall contributions slightly limited.

---

> ### Author Response · Authors · 2023-07-06
> **Response to Reviewer 3EKA (Part 1)**
>
> We thank the reviewer for his or her feedback. Please find some detailed responses to review points below, as well as a revised manuscript with tracked changes in red.
>
> **1. On Policy $P_k$**
>
> In SGD methods, the "policy" or "data sampling distribution" refers to the approach used to select or sample the data points from the training dataset and compute the stochastic gradients during the optimization process. It determines the order and frequency with which the data points are presented to the algorithm for updating the model parameters. The choice of this data sampling strategy in SGD is thus crucial as it can impact the convergence speed, generalization performance, and efficiency of the optimization algorithm.
> While the most classical approaches rely on uniform sampling and mini-batch sampling, it may be more efficient to use more advanced selection sampling strategy such as stratified sampling or importance sampling which assigns different weights to individual data points during the sampling process.
> Indeed, the choice of policy or data sampling strategy depends on various factors, including the dataset size, computational resources, desired convergence speed, and specific characteristics of the learning task. The objective is to strike a balance between exploration (encountering diverse data points) and exploitation (updating the model parameters based on sampled data) to achieve optimal learning and generalization. Different strategies may be more suitable for different datasets or learning tasks, and experimentation and analysis are essential for making informed choices regarding the policy or data sampling strategy in SGD methods.
>
> In Example 1, we present the uniform sampling strategy where $P_k = \sum_{i=1}^n \delta_{z_i}/n$ is constant at each iteration $k$ as well as adaptive (non-uniform) sampling $P_k = \sum_{i=1}^n w_i^{(k)} \delta_{z_i}$ with weights satisfying $\sum_{i=1}^n w_i^{(k)} = 1$ for each $k \geq 0$.
>
> In Example 2, which in the revised version has been refined and has been developed further in an independent section, the policy that is used to generate the gradients evolves through time. In that particular type of sampling problems (as well as in Example 3), using a fixed policy would decrease the efficiency of the algorithm.
>
> → In the revised version, we have added some sentences and remarks in the examples to make this clear.
>
> 2. **Discussion of assumptions**
>
> - **On Learning rate**
>
> For the theory to work, the choice of learning rate in stochastic gradient descent (SGD) analysis can be done in two ways: using a fixed step size (implying convergence to a distribution) or a decaying sequence that satisfies the Robbins-Monro conditions (implying convergence to a stationary point). The two approaches are rather different in spirit as well as the tools they involve in their respective analysis. Our work focuses on the Robbins-Monro type of learning rate. In practice, the choice of learning rate is often determined through experimentation and fine-tuning to achieve the best performance on the given task. It is therefore difficult to cover by the theory.
> - **On Assumption A3**
>
> In the analysis of SGD methods, the requirement for the Hessian matrix evaluated at the optimal point to be positive definite is often needed for stability in the proof of the weak convergence. The positive definiteness of the Hessian matrix provides stability and robustness guarantees in the optimization process. It ensures that small perturbations or noise in the objective function or the training data do not significantly affect the convergence behavior. The positive curvature helps in confining the optimization trajectory near the minimum and prevents it from getting trapped in flat regions or saddle points.
> - **On assumption A5**
>
> Note that Assumptions A2-A5 are stated in the spirit of (Pelletier,1998b) making them mild and general. In particular, Assumptions A4 and A5 are similar to (A1.2) in (Pelletier,1998b) and are *satisfied in most usual cases, e.g.  Robbins-Monro and Kiefer-Wolfowitz algorithms.* More precisely, Assumption A4 is needed to identify the limiting distribution while  Assumption A5 is a stability condition often referred to as the Lyapunov condition. This last condition is technical but not that strong as it is similar to Lindeberg's condition which is necessary  (Hall & Heyde, 1980) for tightness.
>
> The bounded variance assumption is enough to get CLT in a more simple framework with independent observation (e.g., standard CLT). It is also enough to obtain convergence analysis (not weak convergence) for SGD under more restrictive framework such as convexity of the objective as in
>
> Moulines, E., & Bach, F. (2011). Non-asymptotic analysis of stochastic approximation algorithms for machine learning. Advances in neural information processing systems, 24.

---

> ### Author Response · Authors · 2023-07-06
> **Response to Reviewer 3EKA (Part 2)**
>
> In our general setup, we believe that (due to the martingale property that is inherent in our proofs), an additional assumption slightly more restrictive than bounded variance is necessary (as it is the case in the Lindeberg CLT). Finally let us conclude by saying that a growth condition on the variance (Assumption 8 in section 3.4) is needed for almost sure convergence. This is a relaxation of the usual bounded variance assumption.
>
> **Comparison to standard SGD assumptions**
>
> The Lyapunov bound and the standard assumptions for stochastic gradient descent (SGD), such as $\mu$-strong convexity and L-smoothness, are complementary and serve different purposes in the analysis of convergence properties.
>
> - $\mu$-Strong Convexity: The assumption of $\mu$-strong convexity states that the objective function being minimized is at least $\mu$-strongly convex. This means that the function is locally curved upward, implying that the difference between any two points is bounded below by a multiple of their squared distance. The strong convexity assumption is a desirable property as it guarantees convergence to the unique optimal solution as well as fast convergence rates.
>
> - L-Smoothness: The assumption of L-smoothness states that the gradients of the objective function are Lipschitz continuous, with the Lipschitz constant denoted by L. This assumption bounds the rate at which the objective function can change and ensures that the gradients do not vary too rapidly. Smoothness assumptions facilitate convergence analysis by providing bounds on the expected behavior of the gradients.
>
> The Lyapunov bound acts on a really different aspect than convexity or L-Smoothness as it is concerned with the randomness of gradients so it really depends on how we generate the gradients. Hence two types of assumptions are difficult to compare.
>
> → In the revised version, all our previous remarks have been included in Section 2.2 where the concerned assumptions are introduced.
>
> **3. Sketch of proof**
>
> Thank you for your feedback. We have re-written the sketch of the proof to make it more helpful (by interpreting our proof as a well-known bias-variance decomposition). We also added the exact reference in Appendix where the proof can be found.
>
> **4. Optimal choice**
>
> As stated in the introduction (§Contributions), the paper's primary outcome is an equicontinuity property, which is valid when the sequence of conditioning matrices $(C_k)$ converges to a positive definite matrix $C$ (without requiring any further conditions on the error structure). Consequently, the impact of the approximation error resulting from the conditioning matrices assumes a secondary role. In practice it means that not much computational effort should be used to estimate the conditioning matrix $C_k$. This finding encompasses a broad spectrum of Conditioned SGD methods, highlighting the applicability and generalizability of the obtained result.
>
> → In the revised version, we added some remarks in the paragraph §Contributions to clarify this point.
>
> **5. Other assumptions**
>
> Generally speaking, Assumptions A6-A9 are of a different nature than Assumptions A2-A5 as they are meant for the almost sure convergence $\theta_k \to \theta^\star$ while Assumptions A3-A5 are stated for stability and robustness in addition to the event $\{ \theta_k \to \theta^\star\}$ in order to prove the weak convergence.
>
> Furthermore, Assumptions A6-A9 are actually general and mild. Assumption A6 on L-smoothness is standard in optimization and Assumption A7 on the existence of a minorant is one of the weakest assumptions that can be imposed to solve a minimization problem. While Assumptions A6 and A7 deal with the objective function $F$, Assumption A8 is of a different nature as it is concerned with the variance of $g$. For this reason, we cannot deduce A8 from A6 and A7. In Lemma 8 in Appendix C.4, the growth condition is obtained through similar assumptions as A6 and A7 but on $f$ (not $F$). For more details on the growth condition, please refer to the discussion paragraph after Assumption A8. Assumption A9 is actually mild: if we already had the convergence of the matrices $C_k \to C$ then we would find the standard Robbins Monro condition $(\sum_k \gamma_k = \infty,\sum_k \gamma_k^2 < \infty)$. However  $C_k \to C$  is itself conditioned by the convergence of the iterates $\theta_k \to \theta^\star$. The key point of A9 is to allow the matrices $C_k$ to be general by asking for a mild technical assumption on the eigenvalues contrary to
>
> Boyer, C., & Godichon-Baggioni, A. (2023). On the asymptotic rate of convergence of stochastic newton algorithms and their weighted averaged versions. Computational Optimization and Applications, 84(3), 921-972.
>
> where a convergence rate for $C_k \to C$ is required.

---

### Review · Reviewer_bN52 · 2023-07-03

**Summary Of Contributions:**

This work investigates the asymptotic behavior of a general class of preconditioned stochastic gradient descent methods. Under certain technical conditions, the authors characterize the asymptotic normality of the iterates under certain technical conditions, and show that using the inverse of Hessian as the conditioner leads to optimal variance.

**Audience:**

Yes

**Claims And Evidence:**

Yes

**Requested Changes:**

See Strengths And Weaknesses.

**Strengths And Weaknesses:**

### Strength:

- Overall the paper is well-written with clear presentation. The results are corroborated with numerical experiments.

- The analysis does not impose additional assumptions on the convergence rate of the preconditioners $\\{C_k\\}_\{k\ge0\}$, and shows that the algorithm would behave in the same way as an oracle version where $\lim_\{k\to\infty\}C_k$ is used instead. This technical novelty allows one to apply the analysis in a more general way and can be of independent interest.

### Weakness:

- In section 3.4, the authors provide some additional convergence results that are helpful in applying the main results, which require the convergence of the iterates in the first place. The discussion is limited to the stationary points, while the main results also require certain conditions involving the final Hessian matrix $H$. Since a number of existing literature has investigated how SGD algorithms escape from saddle points, it would be helpful if the authors can provide a more detailed discussion on dealing with $H$ in a non-convex setting.

- Assumption 9 introduces a set of extended Robbins-Monro conditions on the preconditioners $\\{C_k\\}_\{k\\ge0\}$, and essentially requires $\\lim_\{k\\to\\infty\} C_k = 0$ (if I am not mistaken), which is not compatible with the conditions in Theorem 2. If this is the case, please make it clear in the discussion.

========

The authors' response has addressed my concerns and I recommend for accept.

---

> ### Author Response · Authors · 2023-07-06
> **Response to Reviewer bN52**
>
> We thank the reviewer for his or her feedback. Please find some detailed responses to review points below, as well as a revised manuscript with tracked changes in red.
>
> **Non-convex setting**
>
> The main goal of the paper is to prove the weak convergence of the rescaled iterates in a non-convex setting for a broad class of conditioning matrices with mild assumptions. First of all we would like to emphasize that if the Hessian matrix at the limiting point is not positive but only semi-definite positive then the limiting distribution would not be the same (as it would be carried over a linear subspace of the ambient space). This is even more difficult when the limiting point is a saddle point. A growing literature considers escaping from saddle points and we believe that the question of weak convergence around these saddle points might be interesting but beyond the scope of the paper.
>
> → In the revised version, this topic is considered in the concluding remarks and we point out several recommendations found in the literature when facing this type of function one of which being the use of a special type of CSGD algorithm as presented in
>
> Antonakopoulos, K., Mertikopoulos, P., Piliouras, G., & Wang, X. (2022, June). AdaGrad avoids saddle points. In International Conference on Machine Learning (pp. 731-771). PMLR.
>
> **Assumption A9**
>
> Assumption 9 provides a control on the eigenvalues of the conditioning matrices $C_k$ to ensure that these preconditioners are well-conditioned. Observe that it does not require or force that the limit $\lim_k C_k$ is null. As a first example, one may consider the case $C_k = Id$ which recovers standard SGD. In this case, the sequences $\mu_k$ and $\nu_k$ are constant equal to $1$ and Assumption 9 boils down to the standard Robbins-Monro condition $\sum_k \gamma_k = +\infty; \sum_k \gamma_k^2 < +\infty$. Then, Assumption 9 may be satisfied in practice for general Conditioned-SGD procedures using the construction of Appendix B.1. By keeping in mind the asymptotic regime $C_k \to C$, the intuition behind Assumption 9 is twofold : one the one hand, the inequality $\mu_k Id \preceq C_{k-1} \preceq \nu_k I_d$ will be satisfied because of the convergence of the preconditioners; on the other hand the condition on the ratio $\nu_k / \mu_k$ guarantees that these sequences are of the same order to avoid ill-conditioning.

---

### Author Response · Authors · 2023-07-06
**General Response to Reviewers**

Dear Reviewers,

We thank you for your constructive feedback. We have just uploaded our revised article - the changes compared to the initial submission are in red. We respond individually to reviews below with our proposed changes. We hope that all of your questions have been answered and are happy to discuss further if needed.

---

### Decision · Action_Editors · 2023-08-11

**Recommendation:** Accept as is

**Comment:**

All reviewers have recommended acceptance, and the authors have already updated their paper addressing the points made by the reviewers.

**Audience:**

Yes.

**Claims And Evidence:**

Yes.